# A Bias Correction Mechanism for Distributed Asynchronous Optimization

**Yuan Gao** *yuan.gao@cispa.de*
*CISPA Helmholtz Center for Information Security*
*Universität des Saarlandes*

**Yuki Takezawa** *yuki-takezawa@ml.ist.i.kyoto-u.ac.jp*
*Kyoto University*
*OIST*

**Sebastian Stich** *stich@cispa.de*
*CISPA Helmholtz Center for Information Security*

**Reviewed on OpenReview:** *https://openreview.net/forum?id=8doMbaah0s*

## Abstract

We develop an asynchronous gradient method for training Machine Learning models with asynchronous distributed workers, each with its own communication and computation pace, and its own local data distribution. In the modern distributed machine learning training process, local data distribution across workers is often heterogeneous (a.k.a. client bias), which is a significant limiting factor in the analysis of most existing distributed asynchronous optimization methods. In this work, we propose AsyncBC, a *distributed asynchronous* variant of the SARAH (Nguyen et al., 2017) method, and show that this is an effective Bias Correction mechanism for distributed asynchronous optimization. We show that AsyncBC can effectively manage arbitrary data heterogeneity, as well as handle gradient updates that arrive in an uncoordinated manner and with delays. As a byproduct of our analysis, we also provide a deeper understanding of the impacts of different stochasticity models on the convergence of the SARAH method.

## 1 INTRODUCTION

Modern machine learning relies heavily on gradient descent and its many variants (Kingma and Ba, 2017). As the size of modern machine learning models grows dramatically, the data required to train such models also becomes huge, making it infeasible to collect all the data on a single machine. Therefore, it is often necessary to perform the training of modern machine learning models (e.g. large language models (Shoeybi et al., 2019), generative models (Ramesh et al., 2021; 2022), and others (Wang et al., 2020)) in a distributed fashion (Bekkerman et al., 2011), where data are distributed across different machines (clients/workers) and updates are coordinated by a parameter server. Another important setting is the federated learning setting (Konečný et al., 2016; Kairouz et al., 2019), where clients (e.g., edge devices or hospitals) jointly train a model without sharing their local data.

Performing gradient methods for large models in this *data-distributed* setting faces many practical challenges, and there is a long line of research attempting to resolve some practical obstacles in distributed training, including communication compression (Seide et al., 2014; Stich et al., 2018; Koloskova et al., 2020a; Fatkhullin et al., 2023; Gao et al., 2024; 2025), decentralization (Lian et al., 2017), local steps (Stich, 2018; Mishchenko et al., 2022c; Jiang et al., 2024a) and their combinations (Condat et al., 2024; Huang et al., 2024). An implicit assumption made in all of these methods is that all workers' computations and communications are synchronized, where all workers and the server wait for the slowest node before starting the next round

of computations. In practice, there are several possible sources for delays, including network latency and hardware heterogeneity (Horvath et al., 2021; Kairouz et al., 2019), and slower "straggler" nodes might significantly hinder the performance of the distributed training process.

A natural approach to mitigate the negative impact of the straggler nodes is the *asynchronous* optimization paradigm. In the asynchronous setting, the server and workers do not wait for the straggler nodes, and when the server receives a gradient from any worker, it immediately takes a step to compute the next point and assigns it to some workers. There is a long line of research studying asynchronous methods (Nedić et al., 2001; McDonald et al., 2010; Agarwal and Duchi, 2011; Arjevani et al., 2020; Stich et al., 2021; Nguyen et al., 2022). Among them, Nguyen et al. (2022) proposed the FedBuff method that demonstrated the practical relevance and superiority of asynchronous methods in the distributed setting that we focus on in this paper. Notably, several recent works attempt to improve the analysis of asynchronous methods in this setting: Koloskova et al. (2022) provided a sharper analysis using the empirical average delay of the workers, Mishchenko et al. (2022a) analyzed the convergence under arbitrary delay patterns, and Islamov et al. (2023) provided a unified analysis of several different asynchronous paradigms.

However, the convergence analysis in these works relies on the bounded gradient dissimilarity assumption, i.e., the distance between the gradient of the local function and the gradient of the global function is upper bounded by some value $\zeta$. The convergence rates of these methods depend on the value of $\zeta$, and therefore their performances would degrade severely when $\zeta$ is large, and when there is no finite value $\zeta$ that satisfies the gradient dissimilarity bound, the algorithm does not converge. In practice, the value of $\zeta$ is typically unknown, making algorithms that depend on this value less robust. The case where gradient dissimilarity is close to 0 is referred to as the data homogeneous setting, but it is often unrealistic in many modern distributed optimization scenarios, especially the cross-device Federated Learning setting (Kairouz et al., 2019; Karimireddy et al., 2020a). To make asynchronous methods applicable to training beyond the data centers where the data is shuffled across all workers, it is crucial to design an asynchronous method that is not affected by data heterogeneity.

## 1.1 Contribution

In this work, we investigate the distributed asynchronous gradient method under arbitrary data heterogeneity.

- We propose and analyze AsyncBC, a distributed asynchronous variant of SARAH, and show that it is an effective bias correction mechanism for distributed asynchronous optimization. While most existing works on distributed asynchronous optimization rely on gradient dissimilarity assumptions to obtain their convergence rate, our method provably converges without any such assumptions, and it is therefore more robust in the practical setting where data heterogeneity might be large.

- We provide convergence analysis of our method in the stochastic setting under a mildly stronger structural assumption of the local functions, and we provide some insights into the necessity of the assumption. As a byproduct, we also demonstrate that SARAH (Nguyen et al., 2017) might fail to converge in the presence of independent stochastic noise.

- We also conduct numerical experiments to corroborate our theoretical findings.

## 1.2 Related Work

**Asynchronous Parallel Optimization.** There is a vast amount of literature in the field of asynchronous optimizations, dating back to the late 1989 (Bertsekas and Tsitsiklis, 2015). Earlier works on asynchronous methods deal with coordinate-wise asynchronicity and the *parallel* setting, with the Hogwild! method being one of the most known works (Recht et al., 2011). There is a line of works extending the methodology of Hogwild!, either improving the analysis (Nguyen et al., 2018) or proposing new variants of the method, including variance-reduction-type variants (Reddi et al., 2015; Zhao and Li, 2016; Mania et al., 2017; Leblond et al., 2017; 2018). These earlier works typically assume that the data is shared across all workers and rely on the sparsity assumption on the local functions (i.e., each local function only affects a small number of

Table 1: Theoretical comparison of our proposed method with asynchronous gradient methods without bias correction. We compare the rates in terms of the number of communications to the server

| Algorithm | BG[a] | BGD[b] | Rate[c] |
|---|---|---|---|
| Async-GD (Koloskova et al., 2022) | No | Yes | $\frac{L_{\max}F_0\sqrt{\tau_C\tau_{\max}}}{T} + \frac{\sqrt{L_{\max}F_0\zeta^2}}{\sqrt{T}} + \left(\frac{L_{\max}F_0\tau_C\zeta}{T}\right)^{2/3}$ |
| AsGrad (Islamov et al., 2023) | Yes | Yes | $\frac{L_{\max}F_0\tau_C}{T} + \frac{\sqrt{L_{\max}F_0\zeta^2}}{\sqrt{T}} + \left(\frac{L_{\max}F_0\tau_C G}{T}\right)^{2/3}$ |
| AsyncBC-GD Ours | No | No | $\frac{L_{\max}F_0\sqrt{\tau_C\tau_{\max}}}{T} + \frac{F_0\sqrt{LL_{\max}\tau_{\max}}}{\sqrt{T}}$ |

(a) **BG** stands for bounded gradient assumption: $\|\nabla f_i(\mathbf{x})\|^2 \leq G^2$ for all $i \in [n]$.

(b) **BGD** stands for bounded gradient dissimilarity assumption: $\|\nabla f_i(\mathbf{x}) - \nabla f_j(\mathbf{x})\|^2 \leq \zeta^2$ for all $i,j \in [n]$.

(c) We present the best-known rates under the most relevant set of assumptions as we use in the analysis. $f$ is assumed to be $L$-smooth and each local function $f_i$ is assumed to be $L_{\max}$ smooth. We omit the initializations that are independent of the target error.

coordinates). In the parallel settings, asynchronicity is typically quantified in terms of bounded overlaps of the update, which is not applicable in the distributed setting.

**Asynchronous Distributed Optimization.** In this work, we do not pursue the parallel direction, which might not fit the practical setting of modern machine learning, where each local function typically affects the entire model, and data is distributed across different workers in a potentially heterogeneous manner (Nguyen et al., 2022). Instead, we consider the distributed optimization setting. In this setting, there is a parameter server that handles the updates, and the asynchronicity is therefore quantified in terms of the delay (or staleness) of the client's state. In recent years, there has been a surge of interest in the distributed setting due to its closer relevance to the modern machine learning practice, especially in the federated learning paradigm (Kairouz et al., 2019). Arjevani et al. (2020) and Stich and Karimireddy (2020) provided the first tight convergence analysis for constant delays, while Koloskova et al. (2022), Mishchenko et al. (2022a), and Islamov et al. (2023) improved various aspects of the analysis of the distributed asynchronous gradient methods. Nguyen et al. (2022) proposed the FedBuff method, which utilized a server buffer to balance the trade-off between the asynchronicity and update quality in practical settings. Such a technique is orthogonal to our method, and might be used in conjunction with our method. Recently, asynchronous reinforcement learning methods have also been studied in (Lan et al., 2024).

In Table 1, we compare our method to recent works on asynchronous gradient methods without bias correction. As is standard in the literature, we measure the complexity of the asynchronous methods in terms of the number of communications on the server side. This is asymptotically equivalent to the total number of finished local gradient oracle calls. We omit the computation of the initial inputs that are independent of the target error.

**Bias Correction.** The seminal work SCAFFOLD of Karimireddy et al. (2020b) introduced the concept of client bias (or client drift) and the idea of bias correction (or drift control) to the distributed and federated learning community. Following SCAFFOLD, there is a large body of works studying the bias correction mechanism for various distributed optimization settings, including distributed or decentralized optimization with communication compression (Mishchenko et al., 2019; Stich, 2020; Richtárik et al., 2021; Gao et al., 2024; Islamov et al., 2024), local updates with infrequent communication (Stich, 2018; Mishchenko et al., 2022b; Jiang et al., 2024a;b), and the combination of these settings (Condat et al., 2023; Grudzień et al., 2023; Huang et al., 2024). Many existing works on bias correction apply global or local control variates that draw inspiration from variance-reduction-type methods (Johnson and Zhang, 2013; Defazio et al., 2014; Nguyen et al., 2017). We are only aware of one concurrent work that studies bias correction in the distributed asynchronous setting (Wang et al., 2025). Their bias correction mechanism is based on the SAG/SAGA style variance reduction methods (Schmidt et al., 2017; Defazio et al., 2014) and is different from ours. We further point out that, while SCAFFOLD style bias correction mechanism is the building block for many existing bias correction methods in other settings, it is not directly applicable to the asynchronous setting, since

SCAFFOLD requires the server to collect full gradients in an epoch-wise manner, which requires epoch-wise synchronizations of the workers.

## 2 PROBLEM FORMULATION

We consider the distributed optimization problem of the form

$$\min_{\mathbf{x} \in \mathbb{R}^d} \left[ f(\mathbf{x}) \coloneqq \frac{1}{n} \sum_{i=1}^n f_i(\mathbf{x}) \right], \tag{1}$$

where the global objective function $f \colon \mathbb{R}^d \to \mathbb{R}$ is defined as the sum of $n$ local objective functions $f_i \colon \mathbb{R}^d \to \mathbb{R}, i \in [n]$. $\mathbf{x}$ represents the parameters of the model. Each worker (client) $i \in [n]$ can only access the local function $f_i$ and its gradient $\nabla f_i$. As is standard in the analysis of non-convex optimization algorithms, we assume that there exists some $f^\star$ such that $f(\mathbf{x}) \geq f^\star$ for all $\mathbf{x} \in \mathbb{R}^d$. The problem (1) covers a wide range of optimization problems arising in the training process of Machine Learning models in distributed (Stich, 2018) or federated (Nguyen et al., 2022) fashion.

We make the following smoothness assumption on the *global* function $f$:

**Assumption 1.** *We say that $f$ is $L$-smooth, i.e. it has $L$-Lipschitz gradient, such that for all $\mathbf{x}, \mathbf{y} \in \mathbb{R}^d$*

$$\|\nabla f(\mathbf{x}) - \nabla f(\mathbf{y})\| \leq L \|\mathbf{x} - \mathbf{y}\| .$$

Note that the smoothness assumption is standard in the non-convex optimization literature. While recent works on asynchronous optimization (Koloskova et al., 2022; Islamov et al., 2023; Mishchenko et al., 2022a) make the stronger assumption that each *local* function $f_i$ is also $L$-smooth, we instead make the following weaker Hessian dissimilarity assumption on the local functions $f_i$:

**Assumption 2.** *We assume that there exists some $\delta > 0$ such that for each $f_i$, we have:*

$$\|\nabla f(\mathbf{x}) - \nabla f(\mathbf{y}) - (\nabla f_i(\mathbf{x}) - \nabla f_i(\mathbf{y}))\| \leq \delta \|\mathbf{x} - \mathbf{y}\| .$$

Note that even though this is called a dissimilarity assumption, this should not be confused as a mere replacement of the usual additional gradient dissimilarity assumption used in prior works (Koloskova et al., 2022; Islamov et al., 2023). Instead, this is a more fine-grained characterization of the $L_i$-smoothness assumption for the local functions, and our results still hold when Assumption 2 is replaced by the usual smoothness assumption of the local functions. In particular, we have the following simple fact:

**Fact 1.** *If each $f_i$ is $L_i$-smooth, then there must exist $0 \leq \delta \leq (L + L_i)$ such that $f$ and $f_i$ satisfies Assumption 2 with parameter $\delta$.*

The proof of Fact 1 is straightforward, and we omit it here. Interested readers can refer to Definition 1 in (Jiang et al., 2024a) or (Khaled and Jin, 2022) and the discussions therein. The main advantage of using this weaker Hessian dissimilarity assumption instead of the usual $L_i$-smoothness assumption for $f_i$ is that it allows us to more clearly characterize the effect of each smoothness parameter. Existing works studied the convergence in terms of $L_{\max} \coloneqq \max_{i \in [n]} L_i$ (by assuming that each $L_i = L_{\max}$). Note that $L \leq \frac{1}{n} \sum_{i \in [n]} L_i$ which is at most, but can potentially be much smaller than, $L_{\max}$.[1] In other words, we have the following chain of inequalities, where the gaps might be large:

$$0 \leq \delta \leq L + L_{\max} \leq 2L_{\max}.$$

By considering the global smoothness and the local smoothness separately via Hessian dissimilarity, we obtain a more fine-grained understanding of the convergence rate's dependency on the smoothness parameters.

Finally, we discuss the assumptions related to asynchronicity and concurrency.

---

[1] Consider the simple case where $f_n(\mathbf{x}) = \frac{a\|\mathbf{x}\|^2}{2}$ and $f_i(\mathbf{x}) = 0, \forall i \in [n-1]. L_{\max}/L = n$.

---

**Algorithm 1** AsyncBC-GD

---

1: **Input:** $\mathbf{x}_0, \mathbf{x}_1, \mathbf{m}_0 = \nabla f(\mathbf{x}_0)$ and concurrency $\tau_{\mathcal{C}}$. Set $\mathbf{x}_1 = \mathbf{x}_0 - \eta \mathbf{m}_0$
2: sever selects u.a.r. a set of active clients of size $\tau_{\mathcal{C}}$ and sends them $\mathbf{x}_1$ and $\mathbf{x}_0$
3:     each active client $k$ computes $\mathbf{g}_1^k = \frac{1}{\tau_{\mathcal{C}}}(\nabla f_k(\mathbf{x}_1) - \nabla f_k(\mathbf{x}_0))$
4: **for** $t = 1, 2, \ldots$ **do**
5:     server receives $\mathbf{g}_{t-\tau_t}^{j_t}$ from client $j_t$
6:     server updates $\mathbf{m}_t = \mathbf{m}_{t-1} + \mathbf{g}_{t-\tau_t}^{j_t}$                              ▷ server updates momentum
7:     server updates $\mathbf{x}_{t+1} = \mathbf{x}_t - \eta \mathbf{m}_t$                                   ▷ server updates parameters
8:     server selects u.a.r. a new client $k_{t+1} \sim [n]$ and sends $\mathbf{x}_{t+1}$ and $\mathbf{x}_t$
9:     client $k_{t+1}$ starts computing $\mathbf{g}_{t+1}^{k_{t+1}} = \nabla f_{k_{t+1}}(\mathbf{x}_{t+1}) - \nabla f_{k_{t+1}}(\mathbf{x}_t)$ ▷ computes next gradient difference

---

**Assumption 3.** *We assume that the staleness of the local states at the clients is at most $\tau_{\max}$. In other words, when a client's output is received by the server at time $t$, the client's state is at a time $t - \tau_t \geq t - \tau_{\max}$.*

The assumption that the local gradients computed by each worker are delayed by at most $\tau_{\max}$ rounds is standard in the distributed asynchronous optimization literature (Stich and Karimireddy, 2020; Nguyen et al., 2022; Koloskova et al., 2022; Islamov et al., 2023). There exist works that do not make any assumptions on the delay pattern (Mishchenko et al., 2022a). However, Mishchenko et al. (2022a) was only able to prove the convergence in the data-homogeneous regime. Under the data-heterogeneous setting without any assumption on the delay pattern, it is fundamentally impossible to obtain any meaningful convergence beyond the gradient dissimilarity: suppose that client 1 **never** responds, then one can only hope to minimize $f - f_1/n$, instead of the true global objective $f$. We also point out that, under certain conditions on the objective function (e.g., variational coherence), one can prove convergence of certain asynchronous methods even when the delay grows polynomially (Zhou et al., 2018).

**Definition 1.** *Throughout the paper, we write $\tau_{\mathcal{C}}$ as the concurrency, the number of active works at each iteration.*

In practice, to best utilize available resources, the concurrency $\tau_{\mathcal{C}}$ would typically be set to $n$, the number of workers. We note that when the concurrency $\tau_{\mathcal{C}}$ equals 1, the asynchronous method reduces to a *synchronized* method. Such a connection might be useful for understanding the limitations of the asynchronous methods. See also our discussion on the basic asynchronous gradient method's dependence on gradient dissimilarity assumptions in the next section.

## 3 BIAS CORRECTION FOR ASYNC OPTIMIZATION

As discussed earlier, the convergence rate of the basic asynchronous gradient method (Koloskova et al., 2022; Islamov et al., 2023) depends on bounded gradient dissimilarity, more precisely, the assumption that $\|\nabla f_i(\mathbf{x}) - \nabla f(\mathbf{x})\|^2 \leq \zeta^2$ for all $i \in [n]$ and $\mathbf{x} \in \mathbb{R}^d$. This is a very restrictive assumption of the objective. We point out that this undesirable dependency on $\zeta$ is not simply a technical deficiency in the analysis, instead, it is a limitation of the method itself. Consider the simplest setting where the number of workers is $n > 1$ while the concurrency is one. Then the basic asynchronous method reduces to SGD for finite sum optimization of batch size one. It is well-known that the convergence of SGD is dependent on $\zeta$ (Lan, 2020; Garrigos and Gower, 2023).[2] Therefore, a bias correction mechanism specialized for distributed asynchronous optimization is necessary to address the algorithm's convergence in the data-heterogeneous regime.

In this section, we propose AsyncBC-GD that provides a bias correction mechanism for asynchronous optimization. At an intuitive level, each local client only contributes the difference between their local gradients at the current point and the previous point, which cancels out the heterogeneity in the local data distribution, while we apply a server-side momentum to ensure sufficient progress. This mechanism is inspired by the SARAH variance reduction mechanism (Nguyen et al., 2017). More precisely, AsyncBC-GD maintains a server-side momentum term $\mathbf{m}_t$. When the server receives the gradients from client $j_t$, whose gradients might

---

[2]The analysis might be improved to only depend on the gradient dissimilarity at the optimum in some cases (Garrigos and Gower, 2023).

be computed at stale points $\mathbf{x}_{t-\tau_t}$ and $\mathbf{x}_{t-\tau_t-1}$, the server updates its momentum term and takes a step in the direction of the new momentum. We highlight the key update steps of AsyncBC-GD below:

$$
\begin{aligned}
\mathbf{m}_t &= \mathbf{m}_{t-1} + \nabla f_{j_t}(\mathbf{x}_{t-\tau_t}) - \nabla f_{j_t}(\mathbf{x}_{t-\tau_t-1}), \\
\mathbf{x}_{t+1} &= \mathbf{x}_t - \eta \mathbf{m}_t.
\end{aligned}
\tag{2}
$$

The server then selects another worker $k_{t+1}$ and sends the updated $\mathbf{x}_{t+1}$ and $\mathbf{x}_t$ to the worker. In this algorithm, the server selects the next worker uniformly at random out of all workers. Such sampling can typically be replaced by a more greedy one where only inactive workers are sampled in the data-homogeneous regime. In the data heterogeneous regime, the greedy sampling might incur a bias error and lead to non-convergence (Mishchenko et al., 2022a; Islamov et al., 2023), but Wang et al. (2025) recently proposed a different approach that claims to support greedy sampling.

We summarize AsyncBC-GD in Algorithm 1. Note that on the client side, at Line-3, we reweight the gradient difference of the initial points. Intuitively, this reweighting at the initial points is important to keep all the computed gradients in balance, since the initial points $\mathbf{x}_1$ and $\mathbf{x}_0$ are the only points at which the gradient differences are computed by $\tau_{\mathcal{C}}$ workers.

**Remark 1** (Communication cost and scalability)**.** *At each iteration of* AsyncBC*, the server receives one message from a client, performs an update and immediately sends two parameter vectors to a chosen client. Therefore the communication cost is not affected by the number of clients n, and scaling up with more clients will not increase the communication workload in each iteration. While as the model size d increases, the cost of communicating a full parameter or gradient vector will increase, there are compression techniques that can be applied to reduce the communication cost (Stich and Karimireddy, 2020; Gao et al., 2024). We leave it for future work to investigate the possibility of applying these techniques to* AsyncBC*. In addition, as the number of clients n increases, the purely asynchronous setting where the server updates the model immediately after receiving a message from a client might not be practical, as the server would have to go through at least n updates before seeing a message from all clients, which might be inefficient as n increases. To address this in practice, we might consider a semi-asynchronous setting where the server buffers some amount of messages from clients before performing one update, similar to the* FedBuff *setting (Nguyen et al., 2022). We also leave it for future work.*

### 3.1 Convergence Analysis of AsyncBC-GD

In this section, we sketch the convergence analysis of Algorithm 1. All missing proofs can be found in the supplement. The main ingredient of our analysis is the virtual iteration technique introduced by Mania et al. (2017). The main difference is that we define the virtual iterates $\widetilde{\mathbf{x}}_t$ in terms of another virtual iterates $\widetilde{\mathbf{m}}_t$, the virtual momentums. The analysis is based on carefully bounding the errors between both the virtual momentum and the true momentum, and the virtual iterate and the true iterate. Variants of virtual iteration techniques that involve multiple virtual iterates have been considered in the literature (Leblond et al., 2018), but our formulation in terms of virtual momentum seems to be novel and might be of independent interest.

But first, it would be handy to also define the set of active workers (and its corresponding timestamp) at each time $t$ precisely: we let $\mathcal{C}_1 := \{(k,1) : k \neq j_1$ is an initial active worker$\}$ be the set of active workers at time 1 (after the first worker $j_1$ has communicated its output), and define $\mathcal{C}_{t+1} := \mathcal{C}_t \setminus \{(j_{t+1}, t+1 - \tau_{t+1})\} \cup \{(k_{t+1}, t+1)\}$ for $t \geq 1$. In the asynchronous optimization literature (Koloskova et al., 2022), the active worker set is often simply defined via the worker only. We use the worker-timestamp notation to more conveniently handle the initialization phase of the algorithm, as the initial states (timestamp 1) are assigned to $\tau_{\mathcal{C}}$ workers. For $t > 1$, there is a one-to-one correspondence between the timestamp $t$ and the assigned worker $k_t$, and we do not need the pair notation.

With the active worker set defined, we can now define our virtual iterate and virtual momentum. We set $\widetilde{\mathbf{x}}_0 := \mathbf{x}_0, \widetilde{\mathbf{x}}_1 := \mathbf{x}_1$ and $\widetilde{\mathbf{m}}_0 := \mathbf{m}_0$ and the following update rules:

$$
\widetilde{\mathbf{x}}_{t+1} := \widetilde{\mathbf{x}}_t - \eta \widetilde{\mathbf{m}}_t, \quad \forall t \geq 1
\tag{3}
$$

and

$$
\begin{aligned}
\widetilde{\mathbf{m}}_1 &:= \widetilde{\mathbf{m}}_0 + \sum_{(k,1) \in \mathcal{C}_1 \cup \{(j_1,1)\}} \mathbf{g}_1^k \\
\widetilde{\mathbf{m}}_t &:= \widetilde{\mathbf{m}}_{t-1} + \mathbf{g}_t^{k_t} \qquad \forall t \geq 2
\end{aligned}
\tag{4}
$$

One of the key limiting factors in existing works (Koloskova et al., 2022; Islamov et al., 2023) is in the upper bound of the errors between the virtual and true iterates, where a gradient dissimilarity assumption or a bounded gradient assumption has to be introduced to control the error. In our work, the error is controlled in terms of local gradient differences, instead of local gradients themselves, and we can therefore bypass the gradient dissimilarity assumption. In particular, we have the following upper bound on the errors:

**Lemma 1.** *Given Assumptions 1 and 2, the sequences $\{\widetilde{\mathbf{x}}_t\}, \{\mathbf{x}_t\}, \{\widetilde{\mathbf{m}}_t\}$ and $\{\mathbf{m}_t\}$ satisfy the following:*

$$
\begin{aligned}
\|\mathbf{m}_t - \widetilde{\mathbf{m}}_t\|^2 &\leq 2\eta^2 \tau_{\mathcal{C}}(\delta^2 + L^2) \sum_{(k,i)\in\mathcal{C}_t} \|\mathbf{m}_{i-1}\|^2, \\
\|\mathbf{x}_t - \widetilde{\mathbf{x}}_t\|^2 &\leq 2\eta^4 (t-1)(\delta^2 + L^2)\tau_{\max}^2 \sum_{i=1}^{t-1} \|\mathbf{m}_{i-1}\|^2.
\end{aligned}
\tag{5}
$$

Note that indeed the upper bound is independent of the data heterogeneity. Now we can state the convergence rate[3] of Algorithm 1:

**Theorem 1.** *Given Assumptions 1 and 2, for the sequence $\{\mathbf{x}_t\}$ generated by Algorithm 1, if $\eta :=$ $\min\left\{\frac{1}{6(\delta+L)\sqrt{\tau_{\mathcal{C}}\tau_{\max}}}, \frac{1}{12\sqrt{L(\delta+L)\tau_{\max}T}}\right\}$, then we have:*

$$
\frac{1}{T}\sum_{t=0}^{T-1} \mathbb{E}\left[\|\nabla f(\mathbf{x}_t)\|^2\right] \leq \mathcal{O}\left(\frac{(\delta+L)\sqrt{\tau_{\mathcal{C}}\tau_{\max}}F_0}{T} + \frac{\sqrt{L(\delta+L)\tau_{\max}}F_0}{\sqrt{T}}\right),
$$

*where $F_0 := f(\mathbf{x}_0) - f^\star$ is the initial suboptimality gap.*

Here we see the advantage of using the $\delta$-Hessian dissimilarity assumption instead of a universal $L_{\max}$-smoothness assumption. The $L_{\max}$ assumption for both the global $f$ and the local $f_i$ made in existing works (Koloskova et al., 2022; Islamov et al., 2023) would lead to $L_{\max}$ dependency in the second term, which might be worse than the $\sqrt{L(\delta+L)}$ dependency that we have now. In the worst case that $L = L_{\max}$ and $\delta = 2L_{\max}$, our convergence rate becomes $\mathcal{O}\left(L_{\max}\sqrt{\tau_{\mathcal{C}}\tau_{\max}}F_0/T + \sqrt{LL_{\max}\tau_{\max}}F_0/\sqrt{T}\right)$.

Now comparing to the best rates known for Async-GD without bias correction (Koloskova et al., 2022) (see Table 1), we see that our method in the worst case obtains the same higher-order term $L_{\max}F_0\sqrt{\tau_{\mathcal{C}}\tau_{\max}}/T$. For the $1/\sqrt{T}$ term, our method obtains $F_0\sqrt{LL_{\max}\tau_{\max}}/\sqrt{T}$ which is independent of the data heterogeneity. On the other hand, the rate in Koloskova et al. (2022) is $\sqrt{L_{\max}F_0\zeta^2}/\sqrt{T}$, which depends on the gradient dissimilarity $\zeta^2$. Note that in the data homogeneous case ($\zeta^2 = 0$), the basic asynchronous gradient descent converges at a $\frac{1}{T}$ rate which is asymptotically faster than ours, while in the data heterogeneous case, our method converges faster when $\zeta^2 \geq LF_0\tau_{\max}$. There also exist simple objectives for which the gradient dissimilarity assumption does not hold for any finite $\zeta$, rendering the basic asynchronous gradient descent non-convergent.[4] While our method is inspired by variance reduction/incremental gradient mechanisms, there exist incremental gradient methods (e.g. SAG (Schmidt et al., 2017) and SAGA (Defazio et al., 2014)) that achieve a $\mathcal{O}(1/T)$ convergence rate in the *synchronized* setting. Our method's $\mathcal{O}(1/\sqrt{T})$ rate seems to be inherent to the algorithm design (as it is also present in the analysis of the synchronized single-loop SARAH method (Nguyen et al., 2017) in the synchronized setting).

We leave for future work to analyze our algorithm in the convex case. Directly transferring our proofs to the convex case seems non-trivial. In particular, it seems unlikely to obtain a tight result by simply plugging our error analysis into a common descent lemma with respect to the primal distance to the optimizer. One can see that, the analysis of SARAH mechanism, the base method underlying our bias-correction mechanism, employs a specific analysis framework in the convex setting. Its convergence is analyzed with respect to the gradient norm, not the usual primal objective gap Nguyen et al. (2017).

---

**Algorithm 2** AsyncBC-SGD

---

1: **Input:** $\mathbf{x}_0, \mathbf{x}_1, \mathbf{m}_0 = \nabla f(\mathbf{x}_0)$ and concurrency $\tau_{\mathcal{C}}$. Set $\mathbf{x}_1 = \mathbf{x}_0 - \eta \mathbf{m}_0$
2: sever selects u.a.r. a set of active clients of size $\tau_{\mathcal{C}}$ and sends them $\mathbf{x}_1$ and $\mathbf{x}_0$
3:     each active client $k$ computes $\mathbf{g}_1^k = \frac{1}{\tau_{\mathcal{C}}}(\nabla f_k(\mathbf{x}_1, \xi_1) - \nabla f_k(\mathbf{x}_0, \xi_1'))^a$
4: **for** $t = 1, 2, \ldots$ **do**
5:     server receives $\mathbf{g}_{t-\tau_t}^{j_t}$ from client $j_t$
6:     server updates $\mathbf{m}_t = \mathbf{m}_{t-1} + \mathbf{g}_{t-\tau_t}^{j_t}$
7:     server updates $\mathbf{x}_{t+1} = \mathbf{x}_t - \eta \mathbf{m}_t$
8:     server selects u.a.r. a new client $k_{t+1} \sim [n]$ and sends $\mathbf{x}_{t+1}$ and $\mathbf{x}_t$
9:     client $k_{t+1}$ starts computing $\mathbf{g}_{t+1}^{k_{t+1}} = \nabla f_{k_{t+1}}(\mathbf{x}_{t+1}, \xi_{t+1}) - \nabla f_{k_{t+1}}(\mathbf{x}_t, \xi_{t+1}')^a$

---

[a] In Section 4.1 we analyze the convergence of the algorithm under the assumption that the stochastic oracle at $\mathbf{x}_{t+1}$ and $\mathbf{x}_t$ share the same randomness $\xi_{t+1}$. This is natural when the randomness comes from sampling a mini-batch of the local data. In most cases, this should be default setup. In Section 4.2 we demonstrate why such an assumption on the randomness is crucial, by constructing a lower bound where the algorithm does not converge when this assumption is violated.

# 4 STOCHASTIC ORACLE AND STOCHASTICITY ASSUMPTIONS

In the previous sections, we mostly considered the deterministic setting, where each client computed the local gradients exactly. The *cross device* Federated Learning setting (Kairouz et al., 2019; Karimireddy et al., 2020a) encompasses a distributed optimization scenario where there might be an extremely large number of clients while each client is resource-poor. In such a setting, the local objectives might be highly heterogeneous, and the local gradients might be computed exactly. However, another important setting of Federated Learning is the *cross-silo* setting, where there is a smaller number of clients that might be resource-rich. In such a case, local gradients are often only approximated stochastically. In this section, we briefly discuss the stochastic variant Algorithm 2 and the stochasticity assumptions that are suitable for our method. As is common in the literature, we will always assume that the stochastic gradients are unbiased:

**Assumption 4.** *For each client $i \in [n]$, the stochastic gradient $\nabla f_i(\mathbf{x}, \xi)$ is unbiased, i.e. $\mathbb{E}\left[\nabla f_i(\mathbf{x}, \xi)\right] = \nabla f_i(\mathbf{x})$.*

## 4.1 Convergence of AsyncBC-SGD with Mean-Squared-Smoothness

In this section, we first show that if we make a slightly stronger structural assumption on the smoothness of the local functions, Algorithm 2 converges. In particular, we introduce the following assumption:

**Assumption 5.** *We say that $f_i(\cdot) = \mathbb{E}_\xi\left[f_i(\cdot, \xi)\right]$ is $\ell$-mean-squared-smooth if:*

$$\mathbb{E}_\xi\left[\|\nabla f_i(\mathbf{x}, \xi) - \nabla f_i(\mathbf{y}, \xi)\|^2\right] \le \ell^2 \|\mathbf{x} - \mathbf{y}\|^2.$$

*Further, we also assume that Algorithm 2 access the stochastic oracle for all pairs $(\mathbf{x}_{t+1}, \mathbf{x}_t)$ with the same randomness $\xi_{t+1}$.*

This assumption is sometimes referred to as mean-squared-smoothness in the literature (Xu and Xu, 2022), and it can also be seen as a stochastic strengthening of the smoothness assumption for each local function. In particular, if for each $\xi$, $f(\mathbf{x}, \xi)$ is $L_{\max}$-smooth, then $\ell \le L_{\max}$. It is a popular stochasticity assumption in the literature of SGD, especially for variance reduction methods (Fang et al., 2018; Cutkosky and Orabona, 2019; Tran-Dinh et al., 2022; Wang et al., 2019; Xu and Xu, 2022). The mean-squared-smoothness assumption relaxes the individual smoothness assumption with respect to each randomness $\xi$. When each local function $f_i$ is a logistic regression loss and each randomness $\xi$ represents a mini-batch of the local data, then $f_i(\cdot, \xi)$ is indivually smooth and $f_i$ is therefore mean-squared-smooth.

In Assumption 5 we also assume that we can sample the local gradient at different points with the same randomness. This is natural when, say, the randomness comes from sampling a mini-batch of the local

---

[3]We omit the initial computation of $\mathbf{m}_0 = \nabla f(\mathbf{x}_0)$, which incurs an additional $n$ term that does not depend on the target error.

[4]Consider $f_i(\mathbf{x}) = i\|\mathbf{x}\|^2/2$.

data. With this assumption, it is easy to have the following convergence statement, independent of any heterogeneity assumption:

**Theorem 2.** *Given Assumptions 1, 4 and 5, for the sequence $\{\mathbf{x}_t\}$ generated by Algorithm 2, if $\eta :=$*
$\min\left\{ \frac{1}{\ell\sqrt{6\tau_C\tau_{\max}}}, \frac{1}{10\ell\sqrt{T}}, \frac{1}{2\sqrt{2L\ell\tau_{\max}(T-1)}} \right\}$, *then we have:*

$$\frac{1}{T} \sum_{t=0}^{T-1} \mathbb{E}\left[ \|\nabla f(\mathbf{x}_t)\|^2 \right] \leq \mathcal{O}\left( \frac{\ell\sqrt{\tau_C\tau_{\max}}F_0}{T} + \frac{(\ell + \sqrt{L\ell\tau_{\max}})F_0}{\sqrt{T}} \right).$$

### 4.2 Unconvergence of AsyncBC-SGD with Independent Noise

In Section 4.1 we showed the convergence of AsyncBC-SGD under Assumption 5. This assumption implies that the noise of the stochastic local gradient is structurally dependent on the local objective. This is natural in many settings, including the popular empirical risk minimization paradigm in most machine learning training processes. In this section, we further illustrate why such a dependence between the noise and the local objective seems necessary for the convergence of AsyncBC-SGD.

In the following, we show that without Assumption 5, if the outputs of each client $i$ are injected with a large independence noise, then the algorithm does not converge even when the concurrency $\tau_C$ is 1. As we have noted in Section 2, when $\tau_C = 1$, the algorithm reduces to a synchronized method. We remark that in this case our Algorithm 2 recovers a stochastic version of SARAH variance reduction method (Nguyen et al., 2017) where for each $f_i$ the gradient difference oracle $\mathbf{g}^i = \nabla f_i(\mathbf{x}, \xi) - \nabla f_i(\mathbf{y}, \xi') = \nabla f_i(\mathbf{x}) - \nabla f_i(\mathbf{y}) + \xi''$ with $\mathbb{E}[\xi''] = \mathbf{0}$ and $\mathbb{E}\left[ \|\xi''\|^2 \right] = \sigma^2$. Note that, unlike Assumption 5, here we do not assume that the oracle accesses at $\mathbf{x}$ and $\mathbf{y}$ need to share the same randomness, but rather, we only quantify the expectation and the variance of $\mathbf{g}^i$. Our next theorem shows that when $\sigma^2 > 0$ Algorithm 2 (and hence SARAH) does not converge to the stationary point.

**Theorem 3** (Lower bound with independent noise). *Consider the single concurrency setting, where $f_i(\mathbf{x}) = \|\mathbf{x}+\mathbf{b}_i\|^2/2$ for any $\mathbf{b}_i$ and $\mathbf{x}_0$ is not a stationary point of $f$. If each $\mathbf{g}_t^i$ output of client $i$ is given by $\mathbf{g}_t^i = \nabla f_i(\mathbf{x}_t) - \nabla f_i(\mathbf{x}_{t-1}) + \xi_t$ where $\xi_t \sim \mathcal{N}(\mathbf{0}, \sigma\mathbf{I})$ is an independent Gaussian noise, then the sequence $\{\mathbf{x}_t\}$ generated by Algorithm 2 does not converge to the stationary point of $f$ for any $\eta < 1$.*

In words, we show that when the gradient difference oracle of Algorithm 2 is injected with independent noise with nonzero variance $\sigma^2 > 0$, then depending on the stepsize, the algorithm either diverges to infinity or does not make enough progress to the stationary point. We verify Theorem 3 with numerical results in Appendix C. In the following we also briefly explain why Theorem 3 does not refute Theorem 2:

**Remark 2.** *We point out that the oracle $\mathbf{g}_t^i$ violates Assumption 5 in the following sense: if $\mathbf{g}_t^i := \nabla f_i(\mathbf{x}_t, \zeta_t) + \nabla f_i(\mathbf{x}_{t-1}, \zeta_t)$ where $\nabla f_i(\mathbf{x}_t, \zeta_t)$ and $\nabla f_i(\mathbf{x}_{t-1}, \zeta_t)$ both satisfy Assumption 5, then the variance of the oracle becomes:*

$$\mathbb{E}_{\zeta_t}\left[ \|\nabla f_i(\mathbf{x}_t, \zeta_t) + \nabla f_i(\mathbf{x}_{t-1}, \zeta_t) - (\nabla f_i(\mathbf{x}_t) - \nabla f_i(\mathbf{x}_{t-1}))\|^2 \right] \leq (L^2 + \ell^2) \|\mathbf{x}_t - \mathbf{x}_{t-1}\|^2.$$

*Therefore, oracles that follow Assumption 5 will have variance dependent on the distance $\|\mathbf{x}_t - \mathbf{x}_{t-1}\|^2$, whereas in Theorem 3 the variance is a constant $\sigma^2$.*

## 5 EXPERIMENTS

In this section, we conduct numerical experiments to validate the theoretical results of our algorithms. All experiments were run on an Intel(R) Xeon(R) CPU E7-8890 v4 @ 2.20GHz chip. All our code can be accessed at here and here. All additional details of the experiments can be found in Appendix D.

**Synthetic Least Squares Problem**  We first consider a simple synthetic least squares problem and verify our algorithm's independence of data heterogeneity. We study the following problem as in (Koloskova et al., 2020b): Each worker $i$ has access to $f_i(\mathbf{x}) := \frac{1}{2} \|\mathbf{A}_i\mathbf{x} - \mathbf{b}_i\|^2$, where $\mathbf{A}_i^2 := \frac{i^2}{n} \cdot \mathbf{I}_d$ and each $\mathbf{b}_i$ is sampled from

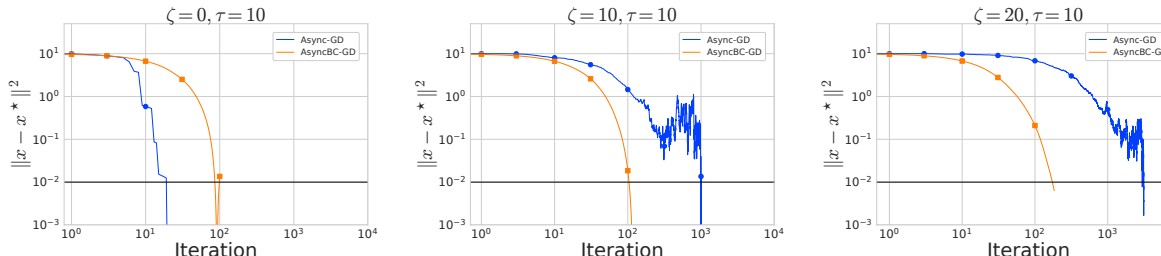

Figure 1: Convergence of AsyncBC-GD and Async-GD under different data heterogeneity. We see that the performance of Async-GD deteriorates as the data heterogeneity increases, while AsyncBC-GD is unaffected.

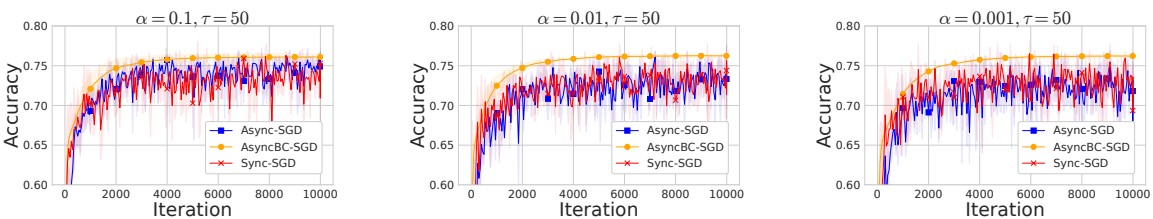

Figure 2: Performance of AsyncBC-SGD, Async-SGD and Sync-SGD for the Fashion MNIST dataset under different data heterogeneity. Decreasing $\alpha$ means increasing data heterogeneity. We see that AsyncBC-SGD's performance is smooth and robust against increasing $\alpha$. It outperforms both Async-SGD and Sync-SGD.

$\mathcal{N}(0, \frac{\zeta^2}{i^2}\mathbf{I}_d)$ for some parameter $\zeta$. It is easy to see that the $\zeta$ parameter controls the gradient dissimilarity (at the optimum[5]). In particular, when $\zeta = 0, \nabla f_i(\mathbf{x}^\star) = 0, \forall i$. The detailed setting of the synthetic experiment can be found in Appendix D.1.

In Figure 1 we investigate the effect of $\zeta$ on the convergence of AsyncBC-GD and Async-GD. To simulate the different computation speeds of the workers, we set each worker $i$'s computation time for each task to $i \times \tau / n$, where $\tau$ controls the asynchronicity. We set $\tau = 10$. We perform a grid search for the best $\eta$ parameter for both algorithms. We observe that AsyncBC-GD's performance is consistent for different $\zeta$ values, as predicted by our theory, while Async-GD takes longer to converge when $\zeta$ increases. Note that in the data homogeneous case ($\zeta = 0$), Async-GD converges faster than AsyncBC-GD, which is consistent with the theory.

**Regularized Logistic Regression For Fashion MNIST Classification**   Now we consider a regularized logistic regression problem for the opensourced Fashion MNIST dataset (Xiao et al., 2017). We use a nonconvex regularizer following (Zhao et al., 2022). The objective function over a pair of data $(\mathbf{a}, b)$ is given as the following:

$$f(\mathbf{x}; (\mathbf{a}, b)) := \log(1 + \exp(-b\mathbf{a}^\top\mathbf{x})) + \rho \sum_{i=1}^{d} \frac{\mathbf{x}_i^2}{1 + \mathbf{x}_i^2},$$

where the last term is a nonconvex regularizer and $\rho = 0.05$ is the regularization parameter. We distribute the training dataset to clients using the Dirichlet distribution with parameter $\alpha$ to control the data heterogeneity (Hsu et al., 2019). In particular, the datasets held by the clients get more heterogeneous as $\alpha$ approaches zero. Additional details can be found in Appendix D.2.

In Figure 2 we summarize the performance of AsyncBC-SGD, Async-SGD and Sync-SGD for the Fashion MNIST dataset. The compare the performance in terms of the number of iterations. We observe that, as data heterogeneity increases, the performance of Async-SGD deteriorates and becomes highly unstable, while AsyncBC-SGD is robust against data heterogeneity and outperforms Async-SGD. Moreover, AsyncBC-SGD

---

[5]This is a somewhat more relaxed notion of gradient dissimilarity than the one considered in (Koloskova et al., 2022; Islamov et al., 2023).

attains a much smoother curve consistently. The stability of the accuracy curve for AsyncBC-SGD verifies the robustness of our method against stochastic noise in a practical setting. We also observer that Sync-SGD is, as expected, unaffected by data heterogeneity. But it is outperformed by AsyncBC-SGD, which might be due to the fact taht AsyncBC-SGD is inherently a variance-reduced method.

## 6 CONCLUSION AND OUTLOOKS

In this work, we investigate the distributed asynchronous gradient methods and address their convergence analysis in the arbitrarily data-heterogeneous setting. We propose and analyze AsyncBC-GD, a distributed asynchronous variant of SARAH. We prove that it is an effective Bias Correction mechanism in that it converges without the restrictive bounded gradient or bounded gradient dissimilarity condition, and is therefore superior in the high data heterogeneous regime than the basic asynchronous methods without bais correction (Nguyen et al., 2022; Koloskova et al., 2022; Mishchenko et al., 2022a; Islamov et al., 2023). We also study the suitable stochasticity assumptions for our method and prove the method's convergence under the mean-squared-smoothness assumption. As a byproduct of our analysis, we also prove the non-convergence of the popular SARAH variance reduction method in the presence of independent gradient noise. Future works can investigate the possibility of obtaining a $\mathcal{O}(1/T)$ rate for distributed asynchronous optimization with either our bias correction mechanism or other methods. Another important theoretical challenge is to further understand the delay dependence of the convergence of bias-corrected asynchronous methods. In particular, future work can investigate the possibility of improving the dependence on $\tau_{\max}$ to $\tau_{\text{avg}}$, or refute such possibility with a lower bound construction.

### Acknowledgments

YT is supported by JSPS KAKENHI Grant Number 23KJ1336.

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

## A  ANALYSIS OF AsyncBC-GD

Recall that we consider the following virtual iterates:

$$\widetilde{\mathbf{x}}_{t+1} := \widetilde{\mathbf{x}}_t - \eta \widetilde{\mathbf{m}}_t, \quad \forall t \geq 1$$

and

$$\begin{aligned}
\widetilde{\mathbf{m}}_1 &:= \widetilde{\mathbf{m}}_0 + \sum_{(k,1)\in\mathcal{C}_1\cup\{j_1\}} \mathbf{g}_1^k \\
\widetilde{\mathbf{m}}_{t+1} &:= \widetilde{\mathbf{m}}_t + \mathbf{g}_t^{k_t} \qquad \forall t \geq 1
\end{aligned}$$

For our convergence analysis, we also consider the following virtual optimality gap:

$$\widetilde{F}_t := \mathbb{E}\left[f(\widetilde{\mathbf{x}}_t) - f^\star\right] \tag{6}$$

We first give a complete analysis of the error of the virtual iterate and virtual momentum:

**Lemma 2.** *Let* $\{\mathbf{x}_t\}, \{\mathbf{m}_t\}, \{\widetilde{\mathbf{x}}_t\}$ *and* $\{\widetilde{\mathbf{m}}_t\}$ *be defined by Algorithm 1,Equation* (3) *and Equation* (4). *Then we have:*

$$\begin{aligned}
\mathbf{m}_t - \widetilde{\mathbf{m}}_t &= - \sum_{(k,i)\in\mathcal{C}_t} g_i^k & \forall t \geq 1 \\
\mathbf{x}_t - \widetilde{\mathbf{x}}_t &= \eta \sum_{i=2}^{t-1} \tau_i^{t-1} \mathbf{g}_i^{k_i} + \eta \sum_{(k,1)\in\mathcal{C}_1} \tau_1^{k,t-1} \mathbf{g}_1^k & \forall t \geq 2
\end{aligned} \tag{7}$$

*where* $\tau_i^t$ *is defined to be the number of times that* $\mathbf{g}_i^{k_i}$ *is delayed until time* $t-1$ *and* $\tau_1^{k,t-1}$ *is the number of times* $\mathbf{g}_1^k$ *is delayed until time* $t-1$.

*Proof.* We first prove the first equation in Equation (7) via induction. By Algorithm 1 and Equation (4), one can easily verify that it holds for $\mathbf{m}_1 - \widetilde{\mathbf{m}}_1$. For $t \geq 2$, we have:

$$\begin{aligned}
\mathbf{m}_t - \widetilde{\mathbf{m}}_t &= \mathbf{m}_{t-1} - \widetilde{\mathbf{m}}_{t-1} + \mathbf{g}_{t-\tau_t}^{j_t} - \mathbf{g}_t^{k_t} \\
&= \left(- \sum_{(k,i)\in\mathcal{C}_{t-1}} \mathbf{g}_i^k\right) + \mathbf{g}_{t-\tau_t}^{j_t} - \mathbf{g}_t^{k_t} = - \sum_{(k,i)\in\mathcal{C}_t} g_i^k,
\end{aligned}$$

where the last equality follows from the definition of $\mathcal{C}_t$.

It remains to prove the error of the virtual iterate. For the second equation in Equation (7), we have for $t \geq 2$:

$$\mathbf{x}_t - \widetilde{\mathbf{x}}_t = -\eta \sum_{s=1}^{t-1} (\mathbf{m}_s - \widetilde{\mathbf{m}}_s)$$

$$= -\eta \sum_{s=1}^{t-1} \sum_{(k,i) \in \mathcal{C}_s} \mathbf{g}_i^k$$

$$= \eta \sum_{i=2}^{t-1} \tau_i^{t-1} \mathbf{g}_i^{k_i} + \eta \sum_{(k,1) \in \mathcal{C}_1} \tau_1^{k,t-1} \mathbf{g}_1^k,$$

where in the last term we treat the $\mathbf{g}_1^k$ and $\mathbf{g}_i^{k_i}, i \geq 2$ separately since the time stamp 1 is the only timestamp that corresponds to multiple workers. $\qquad \square$

Now we upper bound the error between the virtual iterates and the true iterates, using Lemma 2:

**Lemma 1.** *Given Assumptions 1 and 2, the sequences $\{\widetilde{\mathbf{x}}_t\}, \{\mathbf{x}_t\}, \{\widetilde{\mathbf{m}}_t\}$ and $\{\mathbf{m}_t\}$ satisfy the following:*

$$\|\mathbf{m}_t - \widetilde{\mathbf{m}}_t\|^2 \leq 2\eta^2 \tau_{\mathcal{C}} (\delta^2 + L^2) \sum_{(k,i) \in \mathcal{C}_t} \|\mathbf{m}_{i-1}\|^2,$$

$$\|\mathbf{x}_t - \widetilde{\mathbf{x}}_t\|^2 \leq 2\eta^4 (t-1)(\delta^2 + L^2) \tau_{\max}^2 \sum_{i=1}^{t-1} \|\mathbf{m}_{i-1}\|^2. \qquad (5)$$

*Proof.* Consider any worker-time pair $(k,t)$ that was active, we first give a simple upper bound on $\|\mathbf{g}_t^k\|^2$. If $t \geq 1$, we have:

$$\|\mathbf{g}_t^k\|^2 = \|\nabla f_k(\mathbf{x}_t) - \nabla f_k(\mathbf{x}_{t-1}) \pm (\nabla f(\mathbf{x}_t) - \nabla f(\mathbf{x}_{t-1}))\|^2$$

$$\overset{(i)}{\leq} 2\|\nabla f_k(\mathbf{x}_t) - \nabla f_k(\mathbf{x}_{t-1}) - (\nabla f(\mathbf{x}_t) - \nabla f(\mathbf{x}_{t-1}))\|^2 + 2\|\nabla f(\mathbf{x}_t) - \nabla f(\mathbf{x}_{t-1})\|^2$$

$$\overset{(ii)}{\leq} 2(\delta^2 + L^2)\|\mathbf{x}_t - \mathbf{x}_{t-1}\|^2,$$

where in $(i)$ we used the Young's inequality and in $(ii)$ we Assumption 1 and Assumption 2. We can prove something similar for $t = 1$, with an improvement of $\frac{1}{\tau_{\mathcal{C}}^2}$: for any $(k,1) \in \mathcal{C}_1$, we have:

$$\|\mathbf{g}_1^k\|^2 \leq \frac{2}{\tau_{\mathcal{C}}^2} (\delta^2 + L^2) \|\mathbf{x}_1 - \mathbf{x}_0\|^2$$

We first upper bound the error $\|\mathbf{m}_t - \widetilde{\mathbf{m}}_t\|^2, \forall t \geq 1$:

$$\|\mathbf{m}_t - \widetilde{\mathbf{m}}_t\|^2 = \left\| \sum_{(k,i) \in \mathcal{C}_t} \mathbf{g}_i^k \right\|^2$$

$$\leq \tau_{\mathcal{C}} \sum_{(k,i) \in \mathcal{C}_t} \|\mathbf{g}_i^k\|^2$$

$$\leq 2\tau_{\mathcal{C}} (\delta^2 + L^2) \sum_{(k,i) \in \mathcal{C}_t} \|\mathbf{x}_i - \mathbf{x}_{i-1}\|^2$$

$$= 2\eta^2 \tau_{\mathcal{C}} (\delta^2 + L^2) \sum_{(k,i) \in \mathcal{C}_t} \|\mathbf{m}_{i-1}\|^2$$

Similarly, we can upper bound $\|\mathbf{x}_t - \widetilde{\mathbf{x}}_t\|^2, \forall t \geq 2$:

$$
\begin{aligned}
\|\mathbf{x}_t - \widetilde{\mathbf{x}}_t\|^2 &= \eta^2 \left\| \sum_{i=2}^{t-1} \tau_i^{t-1} \mathbf{g}_i^{k_i} + \sum_{(k,1)\in\mathcal{C}_1} \tau_1^{k,t-1} \mathbf{g}_1^k \right\| \\
&\leq \eta^2(t-1)\sum_{i=2}^{t-1}(\tau_i^{t-1})^2 \left\|\mathbf{g}_i^{k_i}\right\|^2 + \eta^2(t-1)(\tau_\mathcal{C}-1)\sum_{(k,1)\in\mathcal{C}_1}(\tau_1^{k,t-1})^2 \left\|\mathbf{g}_1^k\right\|^2 \\
&\leq 2\eta^2(t-1)\sum_{i=2}^{t-1}(\tau_i^{t-1})^2(\delta^2+L^2)\left\|\mathbf{x}_i - \mathbf{x}_{i-1}\right\|^2 \\
&\quad + 2\eta^2(t-1)(\tau_\mathcal{C}-1)^2\sum_{(k,1)\in\mathcal{C}_1}\frac{(\tau_1^{k,t-1})^2}{\tau_\mathcal{C}^2}(\delta^2+L^2)\left\|\mathbf{x}_1 - \mathbf{x}_0\right\|^2 \\
&\leq 2\eta^4(t-1)(\delta^2+L^2)\sum_{i=2}^{t-1}(\tau_i^{t-1})^2\left\|\mathbf{m}_{i-1}\right\|^2 + 2\eta^4(t-1)(\delta^2+L^2)\sum_{(k,1)\in\mathcal{C}_1}\frac{(\tau_1^{k,t-1})^2}{\tau_\mathcal{C}}\left\|\mathbf{m}_0\right\|^2 \\
&\leq 2\eta^4(t-1)(\delta^2+L^2)\tau_{\max}^2\sum_{i=1}^{t-1}\left\|\mathbf{m}_{i-1}\right\|^2
\end{aligned}
$$

$\square$

We can also give a simple corollary of Lemma 1 that lower bounds $\|\widetilde{\mathbf{m}}_t\|^2$ in terms of $\|\mathbf{m}_t\|^2$'s

**Corollary 1.** *Given Assumption 1 and Assumption 2, we have:*

$$
\|\widetilde{\mathbf{m}}_t\|^2 \geq \frac{\|\mathbf{m}_t\|^2}{2} - 2\eta^2\tau_\mathcal{C}(\delta^2+L^2)\sum_{i\in\mathcal{C}_t}\|\mathbf{m}_{i-1}\|^2
$$

This is a simple application of Lemma 1 and Young's inequality and we omit the proof here.

Next we get a descent lemma on the descent of $\widetilde{F}_t$

**Lemma 3.** *Given Assumption 1, the sequences $\{\widetilde{\mathbf{x}}_t\}, \{\mathbf{x}_t\}, \{\widetilde{\mathbf{m}}_t\}$ and $\{\mathbf{m}_t\}$ satisfy the following:*

$$
\widetilde{F}_{t+1} \leq \widetilde{F}_t - \frac{\eta}{2}\mathbb{E}\left[\|\nabla f(\mathbf{x}_t)\|^2\right] - (\frac{\eta}{4} - \frac{L\eta^2}{2})\mathbb{E}\left[\|\widetilde{\mathbf{m}}_t\|^2\right] + \frac{\eta}{2}\mathbb{E}\left[\|\nabla f(\mathbf{x}_t) - \widetilde{\mathbf{m}}_t\|^2\right] + \eta L^2\mathbb{E}\left[\|\widetilde{\mathbf{x}}_t - \mathbf{x}_t\|^2\right] \quad (8)
$$

*Proof.* By Assumption 1, we have:

$$
\begin{aligned}
f(\widetilde{\mathbf{x}}_{t+1}) &\overset{(i)}{\leq} f(\widetilde{\mathbf{x}}_t) - \eta\langle\nabla f(\widetilde{\mathbf{x}}_t), \widetilde{\mathbf{m}}_t\rangle + \frac{L\eta^2}{2}\|\widetilde{\mathbf{m}}_t\|^2 \\
&= f(\widetilde{\mathbf{x}}_t) - \eta\langle\nabla f(\mathbf{x}_t), \widetilde{\mathbf{m}}_t\rangle + \eta\langle\nabla f(\mathbf{x}_t) - \nabla f(\widetilde{\mathbf{x}}_t), \widetilde{\mathbf{m}}_t\rangle + \frac{L\eta^2}{2}\|\widetilde{\mathbf{m}}_t\|^2 \\
&= f(\widetilde{\mathbf{x}}_t) - \frac{\eta}{2}\|\nabla f(\mathbf{x}_t)\| - \frac{\eta}{2}\|\widetilde{\mathbf{m}}_t\|^2 + \frac{\eta}{2}\|\nabla f(\mathbf{x}_t) - \widetilde{\mathbf{m}}_t\|^2 \\
&\quad + \eta\langle\nabla f(\mathbf{x}_t) - \nabla f(\widetilde{\mathbf{x}}_t), \widetilde{\mathbf{m}}_t\rangle + \frac{L\eta^2}{2}\|\widetilde{\mathbf{m}}_t\|^2 \\
&\overset{(ii)}{\leq} f(\widetilde{\mathbf{x}}_t) - \frac{\eta}{2}\|\nabla f(\mathbf{x}_t)\| - \frac{\eta}{2}\|\widetilde{\mathbf{m}}_t\|^2 + \frac{\eta}{2}\|\nabla f(\mathbf{x}_t) - \widetilde{\mathbf{m}}_t\|^2 \\
&\quad + \eta\|\nabla f(\mathbf{x}_t) - \nabla f(\widetilde{\mathbf{x}}_t)\|^2 + \frac{\eta}{4}\|\widetilde{\mathbf{m}}_t\|^2 + \frac{L\eta^2}{2}\|\widetilde{\mathbf{m}}_t\|^2 \\
&= f(\widetilde{\mathbf{x}}_t) - \frac{\eta}{2}\|\nabla f(\mathbf{x}_t)\|^2 - (\frac{\eta}{4} - \frac{L\eta^2}{2})\|\widetilde{\mathbf{m}}_t\|^2 + \frac{\eta}{2}\|\nabla f(\mathbf{x}_t) - \widetilde{\mathbf{m}}_t\|^2 + \eta\|\nabla f(\mathbf{x}_t) - \nabla f(\widetilde{\mathbf{x}}_t)\|^2 \\
&\overset{(iii)}{\leq} f(\widetilde{\mathbf{x}}_t) - \frac{\eta}{2}\|\nabla f(\mathbf{x}_t)\|^2 - (\frac{\eta}{4} - \frac{L\eta^2}{2})\|\widetilde{\mathbf{m}}_t\|^2 + \frac{\eta}{2}\|\nabla f(\mathbf{x}_t) - \widetilde{\mathbf{m}}_t\|^2 + \eta L^2\|\widetilde{\mathbf{x}}_t - \mathbf{x}_t\|^2,
\end{aligned}
$$

where in $(i)$ we used Assumption 1, in $(ii)$ we used Young's inequality, and in $(iii)$ we used Assumption 1 again. Subtracking $f^\star$ and taking expectation on both sides, we get the desired results. $\qquad\square$

Next we bound the gradient error term $\mathbb{E}\left[\|\nabla f(\mathbf{x}_t) - \widetilde{\mathbf{m}}_t\|^2\right]$ recursively

**Lemma 4.** *Given Assumption 2, for all $t \geq 1$, the sequences $\{\mathbf{x}_t\}, \{\widetilde{\mathbf{m}}_t\}$ and $\{\mathbf{m}_t\}$ satisfy the following:*

$$\mathbb{E}\left[\|\nabla f(\mathbf{x}_t) - \widetilde{\mathbf{m}}_t\|^2\right] \leq \mathbb{E}\left[\|\nabla f(\mathbf{x}_{t-1}) - \widetilde{\mathbf{m}}_{t-1}\|^2\right] + \eta^2\delta^2\mathbb{E}\left[\|\mathbf{m}_{t-1}\|^2\right] \tag{9}$$

*Proof.* We first consider the case $t > 1$. Observe that, since $k_t$ is sampled uniformly at random, we have:

$$\mathbb{E}\left[\nabla f(\mathbf{x}_t) - \nabla f(\mathbf{x}_{t-1}) - (\nabla f_{k_t}(\mathbf{x}_t) - \nabla f_{k_t}(\mathbf{x}_{t-1}))\right] = \mathbf{0}$$

Therefore:

$$\begin{aligned}
\mathbb{E}\left[\|\nabla f(\mathbf{x}_t) - \widetilde{\mathbf{m}}_t\|^2\right] &= \mathbb{E}\left[\|\nabla f(\mathbf{x}_t) - \widetilde{\mathbf{m}}_{t-1} - (\nabla f_{k_t}(\mathbf{x}_t) - \nabla f_{k_t}(\mathbf{x}_{t-1}))\|^2\right] \\
&= \mathbb{E}\left[\|\nabla f(\mathbf{x}_{t-1}) - \widetilde{\mathbf{m}}_{t-1} + (\nabla f(\mathbf{x}_t) - \nabla f(\mathbf{x}_{t-1})) - (\nabla f_{k_t}(\mathbf{x}_t) - \nabla f_{k_t}(\mathbf{x}_{t-1}))\|^2\right] \\
&= \mathbb{E}\left[\|\nabla f(\mathbf{x}_{t-1}) - \widetilde{\mathbf{m}}_{t-1}\|^2\right] + \mathbb{E}\left[\|\nabla f(\mathbf{x}_t) - \nabla f(\mathbf{x}_{t-1}) - (\nabla f_{k_t}(\mathbf{x}_t) - \nabla f_{k_t}(\mathbf{x}_{t-1}))\|^2\right] \\
&\quad + \mathbb{E}\left[\langle \nabla f(\mathbf{x}_{t-1}) - \widetilde{\mathbf{m}}_{t-1}, \nabla f(\mathbf{x}_t) - \nabla f(\mathbf{x}_{t-1}) - (\nabla f_{k_t}(\mathbf{x}_t) - \nabla f_{k_t}(\mathbf{x}_{t-1}))\rangle\right] \\
&= \mathbb{E}\left[\|\nabla f(\mathbf{x}_{t-1}) - \widetilde{\mathbf{m}}_{t-1}\|^2\right] + \mathbb{E}\left[\|\nabla f(\mathbf{x}_t) - \nabla f(\mathbf{x}_{t-1}) - (\nabla f_{k_t}(\mathbf{x}_t) - \nabla f_{k_t}(\mathbf{x}_{t-1}))\|^2\right] \\
&\stackrel{(i)}{=} \mathbb{E}\left[\|\nabla f(\mathbf{x}_{t-1}) - \widetilde{\mathbf{m}}_{t-1}\|^2\right] + \delta^2\mathbb{E}\left[\|\mathbf{x}_t - \mathbf{x}_{t-1}\|^2\right] \\
&= \mathbb{E}\left[\|\nabla f(\mathbf{x}_{t-1}) - \widetilde{\mathbf{m}}_{t-1}\|^2\right] + \eta^2\delta^2\mathbb{E}\left[\|\mathbf{m}_{t-1}\|^2\right],
\end{aligned}$$

where in (i) we used Assumption 2. The case $t = 1$ can be handled similarly with a slightly improved bound, but the looser bound is sufficient for our purpose. However, we note that a key point in the proof is the unbiasedness of $\widetilde{\mathbf{m}}_1 - \widetilde{\mathbf{m}}_0$, which is guaranteed by the reweighting of the initial gradient differences in Algorithm 1. $\qquad\square$

**Corollary 2.** *Given Assumption 2, for all $t \geq 1$, the sequences $\{\mathbf{x}_t\}, \{\widetilde{\mathbf{m}}_t\}$ and $\{\mathbf{m}_t\}$ satisfy the following:*

$$\mathbb{E}\left[\|\nabla f(\mathbf{x}_t) - \widetilde{\mathbf{m}}_t\|^2\right] \leq \sum_{i=1}^{t} \eta^2\delta^2\mathbb{E}\left[\|\mathbf{m}_{i-1}\|^2\right]$$

*Proof.* We sum over 1 to $t$ on both sides of Equation (9) and get:

$$\begin{aligned}
\mathbb{E}\left[\|\nabla f(\mathbf{x}_t) - \widetilde{\mathbf{m}}_t\|^2\right] &\leq \mathbb{E}\left[\|\nabla f(\mathbf{x}_0) - \widetilde{\mathbf{m}}_0\|^2\right] + \sum_{i=1}^{t} \eta^2\delta^2\mathbb{E}\left[\|\mathbf{m}_{i-1}\|^2\right] \\
&\stackrel{(i)}{=} \sum_{i=1}^{t} \eta^2\delta^2\mathbb{E}\left[\|\mathbf{m}_{i-1}\|^2\right],
\end{aligned}$$

where for $(i)$ we used the fact that $\widetilde{\mathbf{m}}_0 = \mathbf{m}_0 = \nabla f(\mathbf{x}_0)$. $\qquad\square$

Now we are ready to derive the convergence rate of Algorithm 1 and prove Theorem 1:

**Theorem 1.** *Given Assumptions 1 and 2, for the sequence $\{\mathbf{x}_t\}$ generated by Algorithm 1, if $\eta \coloneqq \min\left\{\frac{1}{6(\delta+L)\sqrt{\tau_\mathcal{C}\tau_{\max}}}, \frac{1}{12\sqrt{L(\delta+L)\tau_{\max}T}}\right\}$, then we have:*

$$\frac{1}{T}\sum_{t=0}^{T-1}\mathbb{E}\left[\|\nabla f(\mathbf{x}_t)\|^2\right] \leq \mathcal{O}\left(\frac{(\delta+L)\sqrt{\tau_\mathcal{C}\tau_{\max}}F_0}{T} + \frac{\sqrt{L(\delta+L)\tau_{\max}}F_0}{\sqrt{T}}\right),$$

*where $F_0 \coloneqq f(\mathbf{x}_0) - f^\star$ is the initial suboptimality gap.*

*Proof.* We plug Lemma 1, Corollary 1, and Corollary 2 into Equation (8) and get for all $t \geq 0$:

$$\widetilde{F}_{t+1} \leq \widetilde{F}_t - \frac{\eta}{2}\mathbb{E}\left[\|\nabla f(\mathbf{x}_t)\|^2\right] - (\frac{\eta}{4} - \frac{L\eta^2}{2})\left(\frac{\mathbb{E}\left[\|\mathbf{m}_t\|^2\right]}{2} - 2\eta^2\tau_{\mathcal{C}}(\delta^2 + L^2)\sum_{i \in \mathcal{C}_t}\mathbb{E}\left[\|\mathbf{m}_{i-1}\|^2\right]\right)$$
$$+ \frac{\eta^3\delta^2}{2}\sum_{i=1}^{t}\mathbb{E}\left[\|\mathbf{m}_{i-1}\|^2\right] + 2\eta^5(t-1)L^2(\delta^2 + L^2)\tau_{\max}^2\sum_{i=1}^{t-1}\|\mathbf{m}_{i-1}\|^2$$

Assume that $\eta \leq \frac{L}{4}$, we get:

$$\widetilde{F}_{t+1} \leq \widetilde{F}_t - \frac{\eta}{2}\mathbb{E}\left[\|\nabla f(\mathbf{x}_t)\|^2\right] - \frac{\eta}{8}\left(\frac{\mathbb{E}\left[\|\mathbf{m}_t\|^2\right]}{2} - 2\eta^2\tau_{\mathcal{C}}(\delta^2 + L^2)\sum_{i \in \mathcal{C}_t}\mathbb{E}\left[\|\mathbf{m}_{i-1}\|^2\right]\right)$$
$$+ \frac{\eta^3\delta^2}{2}\sum_{i=1}^{t}\mathbb{E}\left[\|\mathbf{m}_{i-1}\|^2\right] + 2\eta^5(t-1)L^2(\delta^2 + L^2)\tau_{\max}^2\sum_{i=1}^{t-1}\|\mathbf{m}_{i-1}\|^2$$

Now we sum over 0 to $T-1$ on both sides and get:

$$\frac{\eta}{2}\sum_{t=0}^{T-1}\mathbb{E}\left[\|\nabla f(\mathbf{x}_t)\|^2\right] \leq \widetilde{F}_0 - \frac{\eta}{16}\sum_{t=0}^{T-1}\mathbb{E}\left[\|\mathbf{m}_t\|^2\right] + \underbrace{\frac{\eta^3\tau_{\mathcal{C}}(\delta^2 + L^2)}{4}\sum_{t=1}^{T-1}\sum_{i \in \mathcal{C}_t}\mathbb{E}\left[\|\mathbf{m}_{i-1}\|^2\right]}_{A}$$
$$+ \underbrace{\frac{\eta^3\delta^2}{2}\sum_{t=1}^{T-1}\sum_{i=1}^{t}\mathbb{E}\left[\|\mathbf{m}_{i-1}\|^2\right]}_{B}$$
$$+ \underbrace{2\eta^5L^2(\delta^2 + L^2)\sum_{t=1}^{T-1}(t-1)\tau_{\max}^2\sum_{i=1}^{t-1}\|\mathbf{m}_{i-1}\|^2}_{C}$$

We first bound the third term $A$ on the right-hand side. Note that each of the $\mathbb{E}\left[\|\mathbf{m}_{i-1}\|^2\right]$ is delayed by at most $\tau_{\max}$ rounds, and hence appears at most $\tau_{\max}$ times in $A$, therefore:

$$A \leq \frac{\eta^3\tau_{\mathcal{C}}\tau_{\max}(\delta^2 + L^2)}{4}\sum_{t=0}^{T-2}\mathbb{E}\left[\|\mathbf{m}_t\|^2\right]$$

We simply bound $B$ by:

$$B \leq \frac{\eta^3\delta^2 T}{2}\sum_{t=0}^{T-2}\mathbb{E}\left[\|\mathbf{m}_t\|^2\right]$$

Similarly, we bound $C$ by:

$$C \leq 2\eta^5 L^2(\delta^2 + L^2)\tau_{\max}^2(T-1)^2\sum_{t=0}^{T-2}\mathbb{E}\left[\|\mathbf{m}_t\|^2\right]$$

Therefore, if

$$\eta \leq \min\left\{\frac{1}{6(\delta + L)\sqrt{\tau_{\mathcal{C}}\tau_{\max}}}, \frac{1}{10\delta\sqrt{T}}, \frac{1}{8\sqrt{L(\delta + L)\tau_{\max}(T-1)}}\right\}$$

Then we must have that $A + B + C \leq \frac{\eta}{16} \sum_{t=0}^{T-1} \mathbb{E}\left[\|\mathbf{m}_t\|^2\right]$. Hence,

$$\frac{1}{T} \sum_{t=0}^{T-1} \mathbb{E}\left[\|\nabla f(\mathbf{x}_t)\|^2\right] \leq \frac{2\widetilde{F}_0}{\eta T} = \frac{2F_0}{\eta T}$$

Note that by Fact 1, it suffices to set

$$\eta = \min\left\{\frac{1}{6(\delta + L)\sqrt{\tau_{\mathcal{C}}\tau_{\max}}}, \frac{1}{12\sqrt{L(\delta + L)\tau_{\max}T}}\right\}$$

For such a choice of $\eta$, we get:

$$\frac{1}{T} \sum_{t=0}^{T-1} \mathbb{E}\left[\|\nabla f(\mathbf{x}_t)\|^2\right] \leq \frac{12(\delta + L)\sqrt{\tau_{\mathcal{C}}\tau_{\max}}F_0}{T} + \frac{24\sqrt{L(\delta + L)\tau_{\max}}F_0}{\sqrt{T}}$$

$\square$

## B  ANALYSIS OF AsyncBC-SGD

The convergence analysis of Algorithm 2 under Assumption 5 is very similar to its deterministic counterpart. We point out that most of the differences are due to the application of Assumption 5. Note that Lemma 2 remains the same.

**Lemma 5.** *Given Assumptions 4 and 5, the sequences $\{\widetilde{\mathbf{x}}_t\}, \{\mathbf{x}_t\}, \{\widetilde{\mathbf{m}}_t\}$ and $\{\mathbf{m}_t\}$ satisfy the following:*

$$\begin{aligned}
\mathbb{E}\left[\|\mathbf{m}_t - \widetilde{\mathbf{m}}_t\|^2\right] &\leq \eta^2 \tau_{\mathcal{C}} \ell^2 \sum_{(k,i)\in\mathcal{C}_t} \mathbb{E}\left[\|\mathbf{m}_{i-1}\|^2\right] \\
\mathbb{E}\left[\|\mathbf{x}_t - \widetilde{\mathbf{x}}_t\|^2\right] &\leq \eta^4(t-1)\ell^2\tau_{\max}^2 \sum_{i=1}^{t-1} \mathbb{E}\left[\|\mathbf{m}_{i-1}\|^2\right]
\end{aligned} \tag{10}$$

*Proof.* The proof of Lemma 5 is very similar to that of Lemma 1, except that we need to use Assumption 5 instead of Assumption 2 to bound $\left\|\mathbf{g}_t^k\right\|^2$:

$$\begin{aligned}
\mathbb{E}\left[\left\|\mathbf{g}_t^k\right\|^2\right] &= \mathbb{E}\left[\|\nabla f_k(\mathbf{x}_t, \xi_t) - \nabla f_k(\mathbf{x}_{t-1}, \xi_t)\|^2\right] \\
&\leq \ell^2 \mathbb{E}\left[\|\mathbf{x}_t - \mathbf{x}_{t-1}\|^2\right] \\
&= \eta^2 \ell^2 \mathbb{E}\left[\|\mathbf{m}_{t-1}\|^2\right]
\end{aligned}$$

Again, we first uppper bound the rror $\mathbb{E}\left[\|\mathbf{m}_t - \widetilde{\mathbf{m}}_t\|^2\right], \forall t \geq 1$:

$$\begin{aligned}
\mathbb{E}\left[\|\mathbf{m}_t - \widetilde{\mathbf{m}}_t\|^2\right] &= \mathbb{E}\left[\left\|\sum_{(k,i)\in\mathcal{C}_t} \mathbf{g}_i^k\right\|^2\right] \\
&\leq \tau_{\mathcal{C}} \sum_{(k,i)\in\mathcal{C}_t} \mathbb{E}\left[\left\|\mathbf{g}_i^k\right\|^2\right] \\
&\leq \tau_{\mathcal{C}}\ell^2 \sum_{(k,i)\in\mathcal{C}_t} \mathbb{E}\left[\|\mathbf{x}_i - \mathbf{x}_{i-1}\|^2\right] \\
&= \eta^2 \tau_{\mathcal{C}}\ell^2 \sum_{(k,i)\in\mathcal{C}_t} \|\mathbf{m}_{i-1}\|^2
\end{aligned}$$

Now for $\mathbb{E}\left[\|\mathbf{x}_t - \widetilde{\mathbf{x}}_t\|^2\right], \forall t \geq 2$:

$$\mathbb{E}\left[\|\mathbf{x}_t - \widetilde{\mathbf{x}}_t\|^2\right] = \eta^2 \mathbb{E}\left[\left\|\sum_{i=2}^{t-1}\tau_i^{t-1}\mathbf{g}_i^{k_i} + \sum_{(k,1)\in\mathcal{C}_1}\tau_1^{k,t-1}\mathbf{g}_1^k\right\|^2\right]$$

$$\leq \eta^2(t-1)\sum_{i=2}^{t-1}(\tau_i^{t-1})^2\mathbb{E}\left[\left\|\mathbf{g}_i^{k_i}\right\|^2\right] + \eta^2(t-1)(\tau_{\mathcal{C}}-1)\sum_{(k,1)\in\mathcal{C}_1}(\tau_1^{k,t-1})^2\mathbb{E}\left[\left\|\mathbf{g}_1^k\right\|^2\right]$$

$$\leq \eta^2(t-1)\sum_{i=2}^{t-1}(\tau_i^{t-1})^2\ell^2\mathbb{E}\left[\|\mathbf{x}_i - \mathbf{x}_{i-1}\|^2\right]$$

$$+ \eta^2(t-1)(\tau_{\mathcal{C}}-1)^2\sum_{(k,1)\in\mathcal{C}_1}\frac{(\tau_1^{k,t-1})^2}{\tau_{\mathcal{C}}^2}\ell^2\mathbb{E}\left[\|\mathbf{x}_1 - \mathbf{x}_0\|^2\right]$$

$$\leq \eta^4(t-1)\ell^2\sum_{i=2}^{t-1}(\tau_i^{t-1})^2\mathbb{E}\left[\|\mathbf{m}_{i-1}\|^2\right] + \eta^4(t-1)\ell^2\sum_{(k,1)\in\mathcal{C}_1}\frac{(\tau_1^{k,t-1})^2}{\tau_{\mathcal{C}}}\mathbb{E}\left[\|\mathbf{m}_0\|^2\right]$$

$$\leq \eta^4(t-1)\ell^2\tau_{\max}^2\sum_{i=1}^{t-1}\mathbb{E}\left[\|\mathbf{m}_{i-1}\|^2\right]$$

$$\square$$

We also have the following simple corollary of Lemma 5 that lower bounds $\mathbb{E}\left[\|\widetilde{\mathbf{m}}_t\|^2\right]$ in terms of $\mathbb{E}\left[\|\mathbf{m}_t\|^2\right]$'s:

**Corollary 3.** *Given Assumption 5, we have for all $t \geq 1$:*

$$\mathbb{E}\left[\|\widetilde{\mathbf{m}}_t\|^2\right] \geq \frac{\mathbb{E}\left[\|\mathbf{m}_t\|^2\right]}{2} - \eta^2\tau_{\mathcal{C}}\ell^2\sum_{i\in\mathcal{C}_t}\mathbb{E}\left[\|\mathbf{m}_{i-1}\|^2\right]$$

*Proof.* Again, this is a simple application of Lemma 5 and Young's inequality and we omit the proof here. $\square$

Note that Lemma 3 still holds, and it remains to give a stochastic version of Lemma 4:

**Lemma 6.** *Given Assumptions 4 and 5, for all $t \geq 1$, the sequences $\{\mathbf{x}_t\}, \{\widetilde{\mathbf{m}}_t\}$ and $\{\mathbf{m}_t\}$ satisfy the following:*

$$\mathbb{E}\left[\|\nabla f(\mathbf{x}_t) - \widetilde{\mathbf{m}}_t\|^2\right] \leq \mathbb{E}\left[\|\nabla f(\mathbf{x}_{t-1}) - \widetilde{\mathbf{m}}_{t-1}\|^2\right] + \eta^2\ell^2\mathbb{E}\left[\|\mathbf{m}_{t-1}\|^2\right] \tag{11}$$

*Proof.*

$$
\begin{aligned}
\mathbb{E}\left[\|\nabla f(\mathbf{x}_t) - \widetilde{\mathbf{m}}_t\|^2\right] &= \mathbb{E}\left[\|\nabla f(\mathbf{x}_t) - \widetilde{\mathbf{m}}_{t-1} - (\nabla f_{k_t}(\mathbf{x}_t, \xi_t) - \nabla f_{k_t}(\mathbf{x}_{t-1}, \xi_t))\|^2\right] \\
&= \mathbb{E}\left[\|\nabla f(\mathbf{x}_{t-1}) - \widetilde{\mathbf{m}}_{t-1} + (\nabla f(\mathbf{x}_t) - \nabla f(\mathbf{x}_{t-1})) - (\nabla f_{k_t}(\mathbf{x}_t) - \nabla f_{k_t}(\mathbf{x}_{t-1}))\|^2\right] \\
&= \mathbb{E}\left[\|\nabla f(\mathbf{x}_{t-1}) - \widetilde{\mathbf{m}}_{t-1}\|^2\right] + \mathbb{E}\left[\|\nabla f(\mathbf{x}_t) - \nabla f(\mathbf{x}_{t-1})\|^2\right] \\
&\quad + \mathbb{E}\left[\|\nabla f_{k_t}(\mathbf{x}_t, \xi_t) - \nabla f_{k_t}(\mathbf{x}_{t-1}, \xi_t)\|^2\right] \\
&\quad + 2\mathbb{E}\left[\langle \nabla f(\mathbf{x}_{t-1}) - \widetilde{\mathbf{m}}_{t-1}, \nabla f(\mathbf{x}_t) - \nabla f(\mathbf{x}_{t-1})\rangle\right] \\
&\quad - 2\mathbb{E}\left[\langle \nabla f(\mathbf{x}_{t-1}) - \widetilde{\mathbf{m}}_{t-1}, \nabla f_{k_t}(\mathbf{x}_t, \xi_t) - \nabla f_{k_t}(\mathbf{x}_{t-1}, \xi_t)\rangle\right] \\
&\quad - 2\mathbb{E}\left[\langle \nabla f(\mathbf{x}_t) - \nabla f(\mathbf{x}_{t-1}), \nabla f_{k_t}(\mathbf{x}_t, \xi_t) - \nabla f_{k_t}(\mathbf{x}_{t-1}, \xi_t)\rangle\right] \\
&\overset{(i)}{=} \mathbb{E}\left[\|\nabla f(\mathbf{x}_{t-1}) - \widetilde{\mathbf{m}}_{t-1}\|^2\right] - \mathbb{E}\left[\|\nabla f(\mathbf{x}_t) - \nabla f(\mathbf{x}_{t-1})\|^2\right] \\
&\quad + \mathbb{E}\left[\|\nabla f_{k_t}(\mathbf{x}_t, \xi_t) - \nabla f_{k_t}(\mathbf{x}_{t-1}, \xi_t)\|^2\right] \\
&\overset{(ii)}{\le} \mathbb{E}\left[\|\nabla f(\mathbf{x}_{t-1}) - \widetilde{\mathbf{m}}_{t-1}\|^2\right] + \ell^2 \mathbb{E}\left[\|\mathbf{x}_t - \mathbf{x}_{t-1}\|^2\right] \\
&= \mathbb{E}\left[\|\nabla f(\mathbf{x}_{t-1}) - \widetilde{\mathbf{m}}_{t-1}\|^2\right] + \eta^2 \ell^2 \mathbb{E}\left[\|\mathbf{m}_{t-1}\|^2\right],
\end{aligned}
$$

where in $(i)$ we used $\mathbb{E}\left[\nabla f_{k_t}(\mathbf{x}_t, \xi_t) - \nabla f_{k_t}(\mathbf{x}_{t-1}, \xi_t)\right] = \nabla f(\mathbf{x}_t) - \nabla f(\mathbf{x}_{t-1})$, and in $(ii)$ we used Assumption 5.
$\square$

Therefore, we also have the following corollary:

**Corollary 4.** *Given Assumptions 4 and 5, for all $t \ge 1$, the sequences $\{\mathbf{x}_t\}, \{\widetilde{\mathbf{m}}_t\}$ and $\{\mathbf{m}_t\}$ satisfy the following:*

$$
\mathbb{E}\left[\|\nabla f(\mathbf{x}_t) - \widetilde{\mathbf{m}}_t\|^2\right] \le \sum_{i=1}^{t} \eta^2 \ell^2 \mathbb{E}\left[\|\mathbf{m}_{i-1}\|^2\right]
$$

Now we can prove the convergence of Algorithm 2 under Assumption 5:

**Theorem 2.** *Given Assumptions 1, 4 and 5, for the sequence $\{\mathbf{x}_t\}$ generated by Algorithm 2, if $\eta :=$ $\min\left\{\frac{1}{\ell\sqrt{6\tau_{\mathcal{C}}\tau_{\max}}}, \frac{1}{10\ell\sqrt{T}}, \frac{1}{2\sqrt{2L\ell\tau_{\max}(T-1)}}\right\}$, then we have:*

$$
\frac{1}{T}\sum_{t=0}^{T-1} \mathbb{E}\left[\|\nabla f(\mathbf{x}_t)\|^2\right] \le \mathcal{O}\left(\frac{\ell\sqrt{\tau_{\mathcal{C}}\tau_{\max}}F_0}{T} + \frac{(\ell + \sqrt{L\ell\tau_{\max}})F_0}{\sqrt{T}}\right).
$$

*Proof.* Now we plug Lemma 5 and corollaries 3 and 4 to Equation (8):

$$
\begin{aligned}
\widetilde{F}_{t+1} &\le \widetilde{F}_t - \frac{\eta}{2}\mathbb{E}\left[\|\nabla f(\mathbf{x}_t)\|^2\right] - \left(\frac{\eta}{4} - \frac{L\eta^2}{2}\right)\left(\frac{\mathbb{E}\left[\|\mathbf{m}_t\|^2\right]}{2} - \eta^2 \tau_{\mathcal{C}}\ell^2 \sum_{i \in \mathcal{C}_t} \mathbb{E}\left[\|\mathbf{m}_{i-1}\|^2\right]\right) \\
&\quad + \frac{\eta^3 \ell^2}{2}\sum_{i=1}^{t} \mathbb{E}\left[\|\mathbf{m}_{i-1}\|^2\right] + \eta^5(t-1)L^2\ell^2\tau_{\max}^2 \sum_{i=1}^{t-1}\|\mathbf{m}_{i-1}\|^2
\end{aligned}
$$

Assume that $\eta \leq \frac{L}{4}$, we get:

$$\widetilde{F}_{t+1} \leq \widetilde{F}_t - \frac{\eta}{2}\mathbb{E}\left[\|\nabla f(\mathbf{x}_t)\|^2\right] - \frac{\eta}{8}\left(\frac{\mathbb{E}\left[\|\mathbf{m}_t\|^2\right]}{2} - \eta^2\tau_{\mathcal{C}}\ell^2\sum_{i\in\mathcal{C}_t}\mathbb{E}\left[\|\mathbf{m}_{i-1}\|^2\right]\right)$$
$$+ \frac{\eta^3\ell^2}{2}\sum_{i=1}^{t}\mathbb{E}\left[\|\mathbf{m}_{i-1}\|^2\right] + \eta^5(t-1)L^2\ell^2\tau_{\max}^2\sum_{i=1}^{t-1}\|\mathbf{m}_{i-1}\|^2$$

Now we sum over $0$ to $T-1$ on both sides and get:

$$\frac{\eta}{2}\sum_{t=0}^{T-1}\mathbb{E}\left[\|\nabla f(\mathbf{x}_t)\|^2\right] \leq \widetilde{F}_0 - \frac{\eta}{16}\sum_{t=0}^{T-1}\mathbb{E}\left[\|\mathbf{m}_t\|^2\right] + \underbrace{\frac{\eta^3\tau_{\mathcal{C}}\ell^2}{8}\sum_{t=1}^{T-1}\sum_{i\in\mathcal{C}_t}\mathbb{E}\left[\|\mathbf{m}_{i-1}\|^2\right]}_{A}$$
$$+ \underbrace{\frac{\eta^3\ell^2}{2}\sum_{t=1}^{T-1}\sum_{i=1}^{t}\mathbb{E}\left[\|\mathbf{m}_{i-1}\|^2\right]}_{B} + \underbrace{\eta^5 L^2\ell^2\sum_{t=1}^{T-1}(t-1)\tau_{\max}^2\sum_{i=1}^{t-1}\|\mathbf{m}_{i-1}\|^2}_{C}$$

We first bound the third term $A$ on the right-hand side. Note that each of the $\mathbb{E}\left[\|\mathbf{m}_{i-1}\|^2\right]$ is delayed by at most $\tau_{\max}$ rounds, and hence appears at most $\tau_{\max}$ times in $A$, therefore:

$$A \leq \frac{\eta^3\tau_{\mathcal{C}}\tau_{\max}\ell^2}{8}\sum_{t=0}^{T-2}\mathbb{E}\left[\|\mathbf{m}_t\|^2\right]$$

We simply bound $B$ by:

$$B \leq \frac{\eta^3\ell^2 T}{2}\sum_{t=0}^{T-2}\mathbb{E}\left[\|\mathbf{m}_t\|^2\right]$$

Similarly, we bound $C$ by:

$$C \leq \eta^5 L^2\ell^2\tau_{\max}^2(T-1)^2\sum_{t=0}^{T-2}\mathbb{E}\left[\|\mathbf{m}_t\|^2\right]$$

Therefore, if

$$\eta \leq \min\left\{\frac{1}{\ell\sqrt{6\tau_{\mathcal{C}}\tau_{\max}}}, \frac{1}{10\ell\sqrt{T}}, \frac{1}{2\sqrt{2L\ell\tau_{\max}(T-1)}}\right\}$$

Then we must have that $A + B + C \leq \frac{\eta}{16}\sum_{t=0}^{T-1}\mathbb{E}\left[\|\mathbf{m}_t\|^2\right]$. Hence,

$$\frac{1}{T}\sum_{t=0}^{T-1}\mathbb{E}\left[\|\nabla f(\mathbf{x}_t)\|^2\right] \leq \frac{2\widetilde{F}_0}{\eta T} = \frac{2F_0}{\eta T}$$

For such a choice of $\eta$, we get:

$$\frac{1}{T}\sum_{t=0}^{T-1}\mathbb{E}\left[\|\nabla f(\mathbf{x}_t)\|^2\right] \leq \frac{2\ell\sqrt{6\tau_{\mathcal{C}}\tau_{\max}}F_0}{T} + \frac{(20\ell + 4\sqrt{2L\ell\tau_{\max}})F_0}{\sqrt{T}}$$

$\square$

Finally, we give the lower bound construction for when the gradient oracles are injected noise with bounded variances:

**Theorem 3** (Lower bound with independent noise). *Consider the single concurrency setting, where $f_i(\mathbf{x}) = \|\mathbf{x}+\mathbf{b}_i\|^2/2$ for any $\mathbf{b}_i$ and $\mathbf{x}_0$ is not a stationary point of $f$. If each $\mathbf{g}_t^i$ output of client $i$ is given by $\mathbf{g}_t^i = \nabla f_i(\mathbf{x}_t) - \nabla f_i(\mathbf{x}_{t-1}) + \xi_t$ where $\xi_t \sim \mathcal{N}(\mathbf{0}, \sigma\mathbf{I})$ is an independent Gaussian noise, then the sequence $\{\mathbf{x}_t\}$ generated by Algorithm 2 does not converge to the stationary point of $f$ for any $\eta < 1$.*

*Proof.* For any $\mathbf{x}_t, \mathbf{x}_{t-1}$, we have that $\nabla f_i(\mathbf{x}_t) - \nabla f_i(\mathbf{x}_{t-1}) = \mathbf{x}_t - \mathbf{x}_{t-1}$. Therefore, $\mathbf{m}_t = \mathbf{m}_0 + \sum_{s=1}^{t} \mathbf{g}_s^{j_s} = \mathbf{m}_0 + \mathbf{x}_t - \mathbf{x}_0 + \sum_{s=1}^{t} \xi_s$.

Now we can write down $\mathbf{x}_{t+1}$ recursively:

$$\mathbf{x}_{t+1} = \mathbf{x}_t - \eta\mathbf{m}_t = (1-\eta)\mathbf{x}_t - \eta(\mathbf{m}_0 - \mathbf{x}_0) - \eta\sum_{s=1}^{t}\xi_s$$

Divide both sides by $(1-\eta)^{t+1}$, we have:

$$\frac{\mathbf{x}_{t+1}}{(1-\eta)^{t+1}} = \frac{\mathbf{x}_t}{(1-\eta)^t} - \frac{\eta(\mathbf{m}_0 - \mathbf{x}_0)}{(1-\eta)^{t+1}} - \eta\sum_{s=1}^{t}\frac{\xi_s}{(1-\eta)^{t+1}}$$

Summing both sides from $t=1$ to $T-1$, we have:

$$\frac{\mathbf{x}_T}{(1-\eta)^T} = \frac{\mathbf{x}_1}{1-\eta} - \eta(\mathbf{m}_0 - \mathbf{x}_0)\sum_{t=1}^{T-1}\frac{1}{(1-\eta)^{t+1}} - \eta\sum_{t=1}^{T-1}\sum_{s=1}^{t}\frac{\xi_s}{(1-\eta)^{t+1}}$$

Now we multiply both sides by $(1-\eta)^T$ and get:

$$\mathbf{x}_T = (1-\eta)^{T-1}\mathbf{x}_1 - \eta(\mathbf{m}_0 - \mathbf{x}_0)\sum_{t=1}^{T-1}(1-\eta)^{T-t-1} - \eta\sum_{t=1}^{T-1}\sum_{s=1}^{t}(1-\eta)^{T-t-1}\xi_s$$

For the last term we have:

$$\sum_{t=1}^{T-1}\sum_{s=1}^{t}(1-\eta)^{T-t-1}\xi_s = \sum_{s=1}^{T-1}\sum_{t=s}^{T-1}(1-\eta)^{T-t-1}\xi_s$$
$$= \sum_{s=1}^{T-1}\frac{1-(1-\eta)^{T-s}}{\eta}\xi_s$$

Now by the above equation and the independence of $\xi_i$, we have:

$$\mathbb{E}\left[\|\mathbf{x}_T\|^2\right] \geq \eta^2\sigma^2\sum_{s=1}^{T-1}\left(\frac{1-(1-\eta)^{T-s}}{\eta}\right)^2$$
$$= \sigma^2\sum_{s=1}^{T-1}(1-(1-\eta)^{T-s})^2$$

For step sizes $\frac{1}{2T} \leq \eta \leq 1$, we have $(1-\eta)^{T-s} \leq \frac{1}{1+(T-s)\eta} \leq \frac{1}{\frac{3}{2}-\frac{s}{2T}}$. For $s \leq \frac{T}{2}$, we have $(1-\eta)^{T-s} \leq \frac{4}{5}$. Therefore,

$$\mathbb{E}\left[\|\mathbf{x}_T\|^2\right] \geq \sigma^2\sum_{s=1}^{\lfloor T/2\rfloor}(1-(1-\eta)^{T-s})^2$$
$$\geq \frac{\sigma^2\lfloor T/2\rfloor}{25}$$

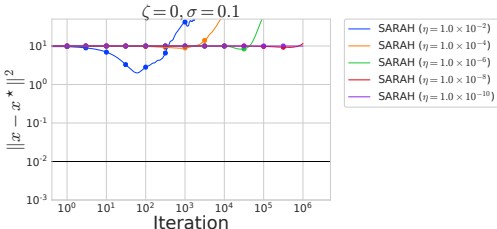

Figure 3: Convergence of SARAH on the synthetic least squares problem with independent noise. We see that the algorithm fails to converge with any of the stepsize choices.

which goes to infinity as $T \to \infty$.

On the other hand, for $\eta \leq \frac{1}{2T}$, we show that even under the data homogeneous ($\mathbf{b}_i = 0$) and noiseless regime ($\sigma^2 = 0$), we cannot reach stationarity if the starting point is not a stationary point. Note that when $\mathbf{b}_i = 0$ and $\sigma^2 = 0$, the only stationary point is $\mathbf{0}$. If $\mathbf{m}_0 = \nabla f(\mathbf{x}_0) = \mathbf{x}_0$, we have $\mathbf{x}_T = (1 - \eta)^{T-1}\mathbf{x}_1 = (1 - \eta)^T \mathbf{x}_0$. Therefore,

$$\|\mathbf{x}_T\|^2 = (1 - \eta)^T \|\mathbf{x}_0\|^2$$
$$\geq (1 - T\eta) \|\mathbf{x}_0\|^2$$
$$\geq \frac{\|\mathbf{x}_0\|^2}{2}$$

Therefore, $\mathbf{x}_T$ cannot converge to $\mathbf{0}$ when $\eta \leq \frac{1}{2T}$. □

## C  SYNTHETIC LEAST SQUARES EXPERIMENT WITH INDEPENDENT NOISE

In this section, we provide an additional experiment verifying Theorem 3. We consider the synthetic least squares problem introduced in Section 5, where we set $n = 4, \zeta = 0$ and concurrency 1, i.e., there is only one active worker. In this case, our algorithm AsyncBC-SGD reduces to SARAH. For each output of the client, we add independent, zero-mean, Gaussian noise with variance $\sigma^2 = 0.01$, i.e., the client outputs $\mathbf{g}_t^i = \nabla f_i(\mathbf{x}_t) - \nabla f_t(\mathbf{x}_{t-1}) + \xi_t$ where $\xi_t \sim \mathcal{N}(\mathbf{0}, \sigma)$. We run the algorithm for stepsizes $\eta \in \{10^{-2}, 10^{-4}, 10^{-6}, 10^{-8}, 10^{-10}\}$ and we see that the algorithm fails to converge for any of the stepsizes. This verifies the lower bound in Theorem 3.

## D  ADDITIONAL EXPERIMENTAL DETAILS

### D.1  Synthetic Least Squares Problem

For the synthetic least squares problem, we set the number of clients $n = 4$, problem dimension $d = 10$, and target error (the average of the last 20 iterations) at 0.01. $\eta$ is searched over $\{1.0 \times 10^{-10}, 5.0 \times 10^{-10}, \cdots, 1.0 \times 10^{-1}\}$.

### D.2  Regularized Logistic Regression For Fashion MNIST Classification

We use 20% of the training dataset for the validation dataset. We set the number of clients $n = 64$, set the batch size to 32, and simulate the different computation speeds as in the previous section, where we set $\tau = 50$. We perform a grid search over $\{0.5, 0.1, 0.05, 0.01, 0.005\}$ for the best $\eta$ parameter, and select the step size with the best average accuracy at the last 10 iterations. For $\alpha = 0.1, 0.01$ and $0.001$, the selected $\eta$ for AsyncBC-SGD are $0.01, 0.005, 0.005$ respectively, and for Async-SGD are $0.01, 0.1, 0.05$ respectively.

