# OpenReview forum: "A Bias Correction Mechanism for Distributed Asynchronous Optimization"
_TMLR — Accepted by TMLR_

### Review · Reviewer_LYnB · 2025-06-05

**Summary Of Contributions:**

This paper proposes AsyncBC, a distributed asynchronous variant of the SARAH algorithm designed to address the challenge of data heterogeneity in asynchronous distributed optimization.

The key contributions are:

1. Bias Correction Without Gradient Dissimilarity Assumption: AsyncBC achieves provable convergence under arbitrary data heterogeneity, removing the common and often impractical assumption of bounded gradient dissimilarity used in prior works.

2. New Convergence Analysis with Virtual Momentum: It introduces a convergence analysis based on a virtual momentum technique, which helps bound the error between true and virtual iterates without requiring assumptions that scale poorly with data heterogeneity.

3. Robustness in Stochastic Settings: The authors extend AsyncBC to stochastic settings (AsyncBC-SGD) and show convergence under the mean-squared-smoothness assumption. They further demonstrate that SARAH-type methods may fail under independent noise, motivating their chosen stochasticity model.

4. Empirical Validation: Experiments on several datasets (Fashion MNIST) show that AsyncBC significantly outperforms baseline asynchronous methods under increasing data heterogeneity, confirming its theoretical robustness and practical effectiveness.

**Audience:**

Yes

**Broader Impact Concerns:**

There is no Broader Impact Concern from me.

**Claims And Evidence:**

Yes

**Requested Changes:**

See Weaknesses 1, 3, 4, and 5. (Weakness 2 is not necessarily required.)

**Strengths And Weaknesses:**

Strengths:

1. Strong Theoretical Contributions.

2. Novel Analytical Technique.

3. Robustness to Data Heterogeneity.

4. Stochastic Analysis and Negative Result.

5. Clear Empirical Validation.

---

Weaknesses:

1. Limited Baselines or Related Work Discussions: The evaluation could be strengthened by including more recent or practical baselines or having some discussions (comparisons) for asynchronous federated optimization, such as FedBuff with drift correction, other SCAFFOLD-inspired variants, and AFedRL with momentum methods [1].

2. Convergence Rate: While the method avoids dependence on gradient dissimilarity, the convergence rate is still $O(1/\sqrt{T})$, which is slower than the $O(1/T)$ rate achievable by some synchronous or homogeneous methods.

3. Assumption Clarification: The mean-squared-smoothness assumption used in the stochastic analysis may not be standard to all readers. A more detailed discussion or examples of its practical relevance would be helpful.

4. Scalability Discussion: While the method targets asynchronous scenarios, the paper does not discuss communication overhead, practical deployment, or scalability across a large number of workers.

5. Typos: Minor typos are present (e.g., "bais correction" instead of "bias correction"), which should be corrected for clarity and polish.

[1] Asynchronous federated reinforcement learning with policy gradient updates: Algorithm design and convergence analysis. ICLR 2025.

---

> ### Author Response · Authors · 2025-06-21
>
> We thank the reviewer for the thoughtful review and we have updated the paper accordingly. Please see below our specific responses to the reviewer's requests.
>
> - We thank the reviewer for pointing out the reference to AFedRL. While it does have an asynchronous element, the method itself is for reinforcement learning, a setting that is different from ours. Nevertheless, we added a brief discussion regarding FedBuff and AFedRL in Section 1.2, paragraph Asynchronous Distributed Optimization. We further added a brief discussion at the end of Section 1.2 regarding SCAFFLOD-style bias correction mechanism and the difficulty of applying it in the context of asynchronous optimization. In particular, we argued that SCAFFOLD-style bias correction requires epoch-wise synchronization of clients because it has to collect the full gradient of all clients periodically to perform the bias correction, which is not possible in the asynchronous setting.
>
> - We leave it for future works to explore if our method can be improved to obtain the $O(1/T)$ rate. The backbone SARAH mechanism requires epoch-wise resynchronization to improve from $O(1/\sqrt(T))$ to $O(1/T)$ [2], so most likely an improvement to our method would also require a more basic improvement to SARAH.
>
> - Following Assumption 5 (the mean-squared-smoothness assumption), we added a brief discussion further explaining this assumption. We point out that this assumption is a relaxation of the individual smoothness with respect to each randomness $\xi$ and it is satisfied in, e.g., a logistic regression setting.
>
> - In Remark 1, Section 3, we added further discussion regarding the scalability of our algorithm. We point out that as the number of clients increases, due to the nature of the asynchronous method, the workload of the server in each iteration does not increase. We leave it for future work to consider the combination of our method with other communication compression mechanisms when the model size increases dramatically. We also briefly discuss a semi-asynchronous setting inspired by FedBuff, which might help improve the trade-off between asynchronicity and practicality when the number of clients is large.
>
> - We thank the reviewer for pointing out the typos, and we will correct them.
>
>
> [2] Nguyen et al, Stochastic Recursive Gradient Algorithm for Nonconvex Optimization (2017)

---

> > ### Comment · Reviewer_LYnB · 2025-08-07
> > **Thanks for the response**
> >
> > Thanks for the explanation. The explanation is satisfactory to me.

---

### Review · Reviewer_dCXw · 2025-07-09

**Summary Of Contributions:**

This paper addresses the challenge of data heterogeneity in distributed asynchronous optimization for large-scale machine learning models. Traditional asynchronous gradient methods often rely on a restrictive bounded gradient dissimilarity assumption, which leads to performance degradation or non-convergence when data distribution across workers is highly heterogeneous. To overcome this, the authors propose AsyncBC, a distributed asynchronous variant of the SARAH method. AsyncBC functions as a bias correction mechanism by having local clients contribute only the difference between their local gradients at current and previous points, effectively canceling out data heterogeneity. The server maintains a momentum term, updated with these gradient differences, and orchestrates the parameter updates. The paper provides theoretical convergence guarantees under a milder mean-squared-smoothness assumption for stochastic settings and empirically validates its performance and robustness against varying data heterogeneity compared to existing asynchronous methods.

**Audience:**

Yes

**Broader Impact Concerns:**

This paper focuses on developing algorithms for distributed computing systems. We do not identify any specific societal risks that need to be highlighted in this context.

**Claims And Evidence:**

Yes

**Requested Changes:**

1. Why is the dominating term in Theorem 1 proportional to $\sqrt{\tau_C}$?

2. The equation $\nabla f_i(x,\xi)-\nabla f_i(y,\xi)=\nabla f_i(x)-\nabla f_i(y)+\xi$ in Sec 4.2 seems to be incorrect. If $x=y$, the left-hand side becomes 0 while the right-hand side is $\xi$. Based on this formulation, the claim in Theorem 3 is problematic.

**Strengths And Weaknesses:**

**Pros:**

- This paper applies a SARAH-style technique to eliminate the variance of client selection. It can be extended to the stochastic gradient setting without a significant modification.

- This paper relaxes the $L$-smooth assumption to a weaker Hessian dissimilarity assumption.


**Cons:**

- The proposed algorithm can not reach comparable performance with Async-GD when the data is homogeneous (Fig 1).

- The Experiments section lacks a comparison with synchronous algorithms.

- In Table 1, the dependence of $F_0$ in the $O(1/T)$ term is $\sqrt{F_0}$ in this work, while the dependence is $F_0$ in existing work.

---

> ### Author Response · Authors · 2025-07-22
>
> We thank the reviewer for their time, and below we clarify and respond to some of the points made by the reviewer:
>
> - Regarding the dependence on $F_0$ in the $O(1/T)$ term in our convergence rate: we point out that our rates depend on $F_0$ linearly, instead of as $\sqrt{F_0}$. This is consistent with Async-GD's and AsGrad's convergence rates.
>
> - Regarding the dependence on the concurrency $\tau_C$: we also clarify that the term that depends on $\tau_C$ is the $O(1/T)$ term, which is not dominating. Nevertheless, such a dependence can also be seen in the convergence analysis for Async-GD (Koloskova et al. 2022) and AsGrad (Islamov et al. 2023) (though for AsGrad, the dependence is $\tau_C$, while for Async-GD and ours, it's $\sqrt{\tau_C \tau_{\max}}$). Technically, the dependence on the concurrency comes from the upper bound on the distance between the actual server-side momentum and the virtual momentum term. The difference between the two is essentially the sum of responses from the current set of active workers. Since the number of active workers is always $\tau_C$, naturally the upper bound would incur a $\tau_C$ dependence. We point to the analysis in Appendix A for more details.
>
> - Regarding the definition of the stochastic oracles in Section 4, we thank the reviewer for catching this. We have updated the paper to solve this issue (marked in green). However, we would like to emphasize that the statement of Theorem 3 is correct, and the updates do not affect the results in Theorem 3. In particular, now we do not directly require that for each pair $(x_{t+1},x_t)$, the stochastic gradient oracles have to use the same randomness in Algorithm 2, and we simply define $g_{t+1}^{k_{t+1}}=\nabla f_{k_{t+1}}(x_{t+1},\xi_{t+1})-\nabla f_{k_{t+1}}(x_{t},\xi_{t+1}')$, emphasizing the potential differences in the randomness by $\xi_{t+1}$ and $\xi_{t+1}'$. In Section 4.1, due to the nature of the mean-squared-smoothness assumption, we further explicitly restrict that $g_{t+1}^{k_{t+1}}=\nabla f_{k_{t+1}}(x_{t+1},\xi_{t+1})-\nabla f_{k_{t+1}}(x_{t},\xi_{t+1})$ in Assumption 5 directly. Now the issue with the definition of $g^i$ in Section 4.2 is easily resolved since the randomness is free to be different, and in Section 4.2 we only quantify the expectation and the variance of the output $g^i$, which we highlight in the update.
>
>   Please let us know if you have any further questions about Section 4 and Theorem 3; we are happy to clarify.

---

> > ### Comment · Reviewer_dCXw · 2025-08-04
> >
> > > Regarding the dependence on $F_0$ in the $O(1/T)$ term in our convergence rate: we point out that our rates depend on $F_0$ linearly, instead of as $\sqrt{F_0}$. This is consistent with Async-GD's and AsGrad's convergence rates.
> >
> > Apology for a typo in my review. I mean, the dependence on $F_0$ in the $O(1/\sqrt{T})$ (the dominant term) is $O(F_0)$ in this paper, while the one in existing work is $O(\sqrt{F_0})$. Could the authors explain why it happens? This inconsistency makes the result incomparable.
> >
> > ---
> > The concerns about $\tau_c$ and the discussion in Sec 4 have been properly addressed. Thanks for your clarification. However, the authors do not address the concerns regarding the comparisons between AsynBC-GD and Asyn-GD, and between AsynBC-GD and other synchronous methods. Although the reviewer can support marginal acceptance, it would render the paper stronger to cover these two comparisons.

---

> > > ### Author Response · Authors · 2025-08-07
> > >
> > > We thank the reviewer for the clarification, and we address the reviewer's further comments in the following:
> > >
> > > - Regarding the $F_0$ factor in the $\frac{1}{\sqrt{T}}$ term:
> > >
> > >   This really touches on the key difference between our bias-corrected method and Async-GD. The dominant term for Async-GD is $\sqrt{\frac{L_{\max}F_0\zeta^2}{T}}$ while ours is $\sqrt{\frac{LL_{\max}F_0^2\tau_{\max}}{T}}$. In place of the $F_0\zeta^2$ factor of Async-GD, our method has $LF_0^2\tau_{\max}$. We note that $\zeta^2$ is defined as the upper bound $\lVert\nabla f_i(x) - \nabla f_j(x)\rVert^2\leq \zeta^2$ for all $i,j\in[n]$. As a sanity check, the squared gradient norm has the same units as $\approx L(F(x)-F^\star)$ (see, for example, Lemma 2.28 in [1] for a demonstration of such a relation in the units), so our rate has the same units as the rate of Async-GD.
> > >
> > >   As we have discussed in the paper, $\zeta^2$ is the gradient dissimilarity upper bound. Our method employs a bias-correction mechanism and its convergence rate is independent of $\zeta^2$. Therefore, in the place of $\zeta^2$, our rate has an additional $LF_0$ factor instead. As we pointed out in the discussions following Theorem 1, our method is faster than Async-GD when $\zeta^2\geq LF_0\tau_{\max}$. Note that there exists simple quadratic objectives where $\zeta^2=\infty$.
> > >
> > > - Regarding the two comparisons:
> > >
> > >   We wish to point out that we do compare the rates between AsynBC-GD and Async-GD in Table 1, and following the discussion after Theorem 1. In the experiment section, we compare the practical performance of these two methods for both the least squares problem and the Fashion MNIST problem.
> > >
> > >   Regarding an experimental comparison between the asynchronous methods and the synchronous methods, we wish to point out that this paper is most theory-focused, and our experiments are run with a simulated asynchronous setup. In such a simulated setup, it would be unfair to compare the performance of asynchronous methods to synchronous methods because of the overhead of the simulation of asynchronicity and distributed computation. On the flip side, we can also easily create a simulation setup where asynchronous methods are arbitrarily faster than synchronous methods. For instance, we can set one client to be as slow as needed, while letting all local objectives be the same. Therefore, to realistically compare asynchronous and synchronous methods, we must consider a realistic distributed training environment. This is, however, beyond the scope of this paper. It would be an interesting future work to perform a comprehensive experimental evaluation of asynchronous methods and synchronous methods in a realistic distributed training setup.
> > >
> > > [1] Garrigos, Guillaume, and Robert M. Gower. "Handbook of convergence theorems for (stochastic) gradient methods." arXiv preprint arXiv:2301.11235 (2023).

---

> > > > ### Comment · Reviewer_dCXw · 2025-09-03
> > > >
> > > > The reviewer thanks the authors for their detailed elaboration.
> > > >
> > > > - Regarding the $F_0$ factor in the $\frac{1}{\sqrt{T}}$ term. This concern has been addressed.
> > > >
> > > > - Regarding the two comparisons. The reviewer understands that it is unfair to compare the synchronous and asynchronous algorithms in the sense of wall-clock time. However, it is still valuable to compare their loss-vs-iteration curves, which reflect their iteration complexity, even though one iteration take different time. Ideally, asynchronous algorithms is sightly slower than synchronous algorithms, while the gap is affordable so that asynchronous parallel computing will compensate it. It is suggested that the authors can compare them in Figure 1 & 2.
> > > >
> > > > Best,
> > > > Reviewer dCXw

---

> > > > > ### Author Response · Authors · 2025-09-30
> > > > >
> > > > > We thank the reviewer for suggesting the experiment comparing against synchronized sgd. We updated the paper to include synchronized sgd. We observed that, even in terms of the number of iterations, AsyncBC-SGD outperforms synchronized sgd, which might be attributed to the fact that AsyncBC-SGD is inherently a variance-reduced method. We marked the discussion in purple in the updated paper.

---

> > > > > > ### Comment · Reviewer_dCXw · 2025-10-12
> > > > > >
> > > > > > Thanks for the clarification from the authors. The reviewer's concern has been addressed.

---

### Review · Reviewer_2Ve8 · 2025-08-04

**Summary Of Contributions:**

The paper studies distributed machine learning problems with distributed asynchronous workers, where each local worker has access to a local data distribution. The authors build on an existing distributed asynchronous algorithm: SARAH, to introduce a bias correction mechanism. With this new mechanism, the authors prove that this variant can converge under data heterogeneity, avoiding classical gradient dissimilarity assumptions.

**Audience:**

Yes

**Claims And Evidence:**

Yes

**Requested Changes:**

See comments in Weaknesses.

**Strengths And Weaknesses:**

Strengths:

- It is important to study realistic distributed optimization settings involving asynchronous workers. The authors provide a new method by incorporating previous gradient difference as a momentum into the update, effectively removing heterogeneous effects.

- Theoretically, the authors prove that the new method converges in deterministic and stochastic settings, under smoothness and dissimilarity assumptions.

Weaknesses:

- Although the focus of this paper is general non-convex problems, it would be helpful if the authors can specialize to convex problems. This can be very useful to evaluate the tightness of the bound in terms of heterogeneous effects and convergence rates.

- The current analysis assumes that the delay is bounded by a constant. This might not be true in some cases like failed workers. It would be useful if the authors could relax this assumption.

- The experiments could benefit by demonstrating the scalability in terms of the number of workers.

---

> ### Author Response · Authors · 2025-08-07
>
> We thank the reviewer for their review and address the comments in the following:
>
> - Convex case:
>
>   It would be very interesting to derive a convergence rate of our algorithm in the convex case. However, directly transferring our proofs to the convex case seems non-trivial. In particular, it seems unlikely to obtain a tight result by simply plugging our error analysis into a common descent lemma with respect to the primal distance to the optimizer. One can see that, the analysis of SARAH mechanism, the base method underlying our bias-correction mechanism, employs a specific analysis framework in the convex setting. Its convergence is analyzed with respect to the gradient norm, not the usual primal objective gap [1]. Therefore, we think that the analysis in the convex setting might be more appropriate for future work.
>
>   In addition, we also note that most recent works on asynchronous optimization focus on the non-convex setting, like our work. See for example, [2,3]
>
> - Bounded delay:
>
>   In this work, we indeed assume that the delay is bounded by $\tau_{\max}$. Such an assumption is particularly important in the heterogeneous setting that we consider. In the discussion following Assumption 3, we pointed out that if the delay is unbounded, i.e. there are workers that never responds, then it is fundamentally impossible to obtain any meaningful convergence beyond the gradient dissimilarity, because at best we can only optimize the objectives of the workers that responded with no access to the local objectives of the workers that failed.
>
> - Scalability:
>
>   Indeed, it would be interesting to see if our algorithm can scale up in practice as the number of workers increases. We point out that one of the benefits of asynchronous methods is that the workload per iteration of the server does not increase as the number of workers increases, because at each iteration, the server only needs to handle the response from one worker. On the flip side, in Remark 1, we pointed out that the server would have to go through at least $n$ iterations before it sees all the workers. This might be inefficient. To address these issues in practice, we might consider a semi-asynchronous setting where the server buffers some amount of messages before performing one update.
>
>   However, our paper is theory-focused and our computational resources are limited. Our experiments are conducted in a simulation setup and increasing $n$ might introduce too much overhead in the simulation. We leave it for future work to conduct a comprehensive study on the scalability of our methods in practice, and verify the performance of our method in a realistic setup.
>
>
> [1] Nguyen, Lam M., et al. "SARAH: A novel method for machine learning problems using stochastic recursive gradient." International conference on machine learning. PMLR, 2017.
>
> [2] Koloskova, Anastasiia, Sebastian U. Stich, and Martin Jaggi. "Sharper convergence guarantees for asynchronous SGD for distributed and federated learning." Advances in Neural Information Processing Systems 35 (2022): 17202-17215.
>
> [3] Islamov, Rustem, Mher Safaryan, and Dan Alistarh. "AsGrad: A sharp unified analysis of asynchronous-SGD algorithms." International Conference on Artificial Intelligence and Statistics. PMLR, 2024.

---

> > ### Comment · Reviewer_2Ve8 · 2025-08-23
> >
> > Thank you for your response. I would recommend also adding comments on the technical difference between convex and nonconvex settings. Regarding the bounded delay, it seems to be possible in some setting as shown by this reference: https://proceedings.mlr.press/v80/zhou18b.html.

---

> > > ### Author Response · Authors · 2025-08-26
> > >
> > > We thank the reviewer for the comment. We have added a further discussion regarding the delay assumption and the work by Zhou et al pointed out by the reviewer. Indeed the work by Zhou et. al. assumed that the objective is variationally coherent. We also added the discussion regarding the convex setting. These updates are marked in pink in our paper.

---

> > > > ### Comment · Reviewer_2Ve8 · 2025-08-26
> > > >
> > > > Thank you for the update. I don't have further comments.

---

### Decision · Action_Editor_NNtH · 2025-10-12

**Recommendation:** Accept as is

**Audience:**

Yes

**Audience Explanation:**

Distributed optimization is central to machine learning and underpins effective practical implementations. So this topic should appeal to a broad TMLR readership.

**Claims And Evidence:**

Yes

**Claims Explanation:**

The paper proposes an asynchronous distributed optimization algorithm that incorporates a bias-correction mechanism, establishes convergence guarantees for the resulting method, and demonstrates the effectiveness of bias correction. Experiments on simple objective functions are provided to validate the theoretical results.